# NK cell spatial dynamics and IgA responses in gut-associated lymphoid tissues during SIV infections

Philippe Rascle [1,2], Cyril Planchais[3,4], Béatrice Jacquelin [1], Marie Lazzerini[1], Vanessa Contreras[5], Caroline Passaes[1], Asier Saez-Cirion [1], Hugo Mouquet[3,4], Nicolas Huot[1] & Michaela Müller-Trutwin [1✉]

HIV infection induces tissue damage including lymph node (LN) fibrosis and intestinal epithelial barrier disruption leading to bacterial translocation and systemic inflammation. Natural hosts of SIV, such as African Green Monkeys (AGM), do not display tissue damage despite high viral load in blood and intestinal mucosa. AGM mount a NK cell-mediated control of SIVagm replication in peripheral LN. We analyzed if NK cells also control SIVagm in mesenteric (mes) LN and if this has an impact on gut humoral responses and the production of IgA known for their anti-inflammatory role in the gut. We show that CXCR5 + NK cell frequencies increase in mesLN upon SIVagm infection and that NK cells migrate into and control viral replication in B cell follicles (BCF) of mesLN. The proportion of IgA+ memory B cells were increased in mesLN during SIVagm infection in contrast to SIVmac infection. Total IgA levels in gut remained normal during SIVagm infection, while strongly decreased in intestine of chronically SIVmac-infected macaques. Our data suggest an indirect impact of NK cell-mediated viral control in mesLN during SIVagm infection on preserved BCF function and IgA production in intestinal tissues.

[1] Institut Pasteur, Unité HIV Inflammation and Persistance, Université de Paris, Paris, France. [2] Université Paris Diderot, Sorbonne Paris Cité, Paris, France. [3] Institut Pasteur, Laboratory of Humoral Immunology, Université de Paris, Paris, France. [4] INSERM U1222, Paris, France. [5] CEA, Université Paris Sud 11, INSERM U1184, Immunology of Viral Infections and Autoimmune Diseases, IDMIT, IBFJ, Fontenay-aux-Roses, France. ✉email: mmuller@pasteur.fr

The immune system evolved to limit the negative effects exerted by pathogens on the host. Infections often result in tissue damage, especially if replication of the pathogen is not controlled[1]. Tissue damage can be triggered directly by pathogens or indirectly by host responses to the infection. Disease tolerance is a defense strategy against tissue damage induced by chronic infection that sustains host homeostasis, without exerting a direct negative impact on pathogens[2]. In people living with HIV (PLWH), anti-retroviral treatment (ART) has transformed a deadly disease into a manageable chronic infection. However, a residual chronic inflammation persists in PLWH who started ART only after several years of infection in chronic infection, which still represents the most frequent case. This chronic inflammation in PLWH under effective ART is associated with the risk of non-AIDS morbidities and mortality[3–5]. One factor that might largely contribute to the persistent inflammation is the disruption of the intestinal barrier and subsequent bacterial translocation in PLWH[6,7].

ART treatment is not able to eliminate the virus, which hides throughout the body in reservoirs. The largest HIV reservoir resides in the intestine[8,9]. Residual viral replication can be observed in follicular helper CD4$^+$ T (T$_{FH}$) cells of B cell follicles (BCF) within lymph nodes (LN) during chronic infection in long-term treated PLWH[10].

Natural hosts of SIV, such as African Green Monkeys (AGMs), do not display chronic inflammation despite stable high viremia during SIV infection[6,11]. Interestingly, SIV-infected AGMs generally do not show any major tissue damage. Thus, LN do not display fibrosis and the follicular dendritic cell (FDC) network within BCF of secondary lymphoid organs is maintained throughout infection in contrast to PLWH[12]. The intestinal epithelial barrier also remains intact[13,14] and no microbial translocation occurs[14–16]. The maintenance of normal LN architecture can be explained by a rapid and strong viral control in this site[17]. Indeed, AGM mount a tissue-specific viral control in secondary lymphoid organs, which is predominantly mediated by NK cells. Thus, SIVagm replication is strongly controlled in LN and spleen with no or little viral replication in BCF, while the virus continues to replicate efficiently in the intestine[18].

The reasons why SIVagm replication in the intestine does not lead to disruption of the intestinal barrier in AGM are unclear. The underlying mechanisms could be multiple. The maintenance of the gut barrier could be related to the preservation of Th17 cells in SIVagm-infected AGM[19]. Immunoglobulins A (IgA) are also known to play an important role in the control of intestinal inflammation[20]. IgA are the dominant antibody isotype found in mucosal secretions[21]. Secretory IgA (SIgA) limit the penetration of commensal bacteria through the epithelium[22] achieving efficient protection of the epithelial barrier by immune exclusion[23]. SIgA exert an anti-inflammatory function in the gut and play a key role in the prevention of tissue damage and recovery from infection[24,25]. IgA-deficient humans indeed exhibit gut microbiota dysbiosis, inflammatory bowel disease and gene expression pattern (i.e., Interferon-stimulate gene expression profiles) similar as in HIV infection[26,27].

B-cell dysregulation, including in mucosa, was described early in the HIV/AIDS epidemics. Damage to the B-cell compartment in HIV infection includes loss of normal BCF architecture in LN, polyclonal hypergammaglobulinemia, increased turnover of B cells, and eventually irreversible loss of memory B-cell responses with advancing HIV disease[28–30]. IgA-producing B cells and plasma cells are not spared from the HIV-induced damage[28]. There are unusually low anti-HIV IgA responses when compared to IgG responses in mucosal fluids[31]. The level of inflammation markers correlates with the loss of IgA in plasma and intestine during HIV infection[29]. PLWH showed dysfunction of B cell isotype switching, leading to limited anti-microbial IgA production[29]. In contrast, AGMs develop higher-magnitude plasma GP120-specific IgA and IgG responses than macaques (MAC, the animal model of HIV), whereas the latter display more robust GP140-directed IgG responses[32,33].

Gut-IgA antibodies are principally produced from plasma cells generated in BCF of Peyer's patches and mesLN. Mesenteric LN draining the intestine amplify IgA responses, but also display high levels of viral replication in SIVmac-infected MAC from early stage of infection on and constitute major viral reservoirs during chronic HIV-1 and SIVmac infection[34]. We raise the hypothesis that the control of SIVagm replication in BCF in AGM could have a beneficial impact on the maintenance of physiological IgA levels and thereby, on regulation of inflammation in the intestine. To address this question, we analyzed NK cells, viral replication, SIgA, and isotype class switch from memory B cells in mesLN and intestine of AGM during SIVagm infection compared to SIVmac-infected macaques. Our study reveals a relation between NK cell-mediated control of SIV infection and IgA levels in the mesLN and delivers innovative insights into the mechanisms that protect the mucosal tissue integrity during lentiviral infections.

## Results

**Induction of CXCR5+ NK cells in B cell follicles from mesenteric lymph nodes during SIVagm infection.** Previous studies have shown that the control of SIVagm replication in secondary lymphoid tissues[12,35] was mediated by NK cells[17]. So far, however, NK cells have only been studied in peripheral LN and spleen during SIVagm infection. No data were available on NK cells in LN draining the intestine in natural hosts of SIV. We thus investigated NK cells in mesLN and first addressed the question whether NK cells from mesLN up-regulate CXCR5 in response to SIVagm infection in AGM, and whether they migrate into BCF of mesLN. The mesLN from uninfected AGMs ($n = 4$) and cynomolgus macaques (MAC, $n = 4$) were compared to mesLN collected during SIVagm ($n = 4$ AGM), and SIVmac infection ($n = 4$ MAC). The viremia levels of the SIV-infected animals are shown in Supplementary Fig. 1a and Supplementary Data 1, 2. The median viremia levels at time of analysis for the chronically infected AGM were $6.56 \times 10^6$ viral RNA copies/mL ($1 \times 10^4 – 1.9 \times 10^6$) and for the chronically infected MAC $3.66 \times 10^6$ viral RNA copies/mL ($3 \times 10^2 – 2.5 \times 10^6$) (Supplementary Fig. 1A, Supplementary Data 1, 2). The viral replication profiles in mesLN were analyzed in situ by FISH. As expected, we did not observe SIV RNA in mesLN from non-infected animals (Fig. 1a). Signs of robust viral production were observed only in mesLN of SIV-infected MAC but not during chronic SIVagm infection (Fig. 1a). These data are in line with previous results showing a viral control in mesLN of SIV-infected AGM[35]. We next investigated the localization of NK cells inside the mesLN by immunohistochemistry. We generally did not observe NK cells in BCF of mesLN from non-infected animals nor from SIV-infected MAC (Fig. 1b). However, NK cells were readily detectable in BCF of mesLN from SIV-infected AGM (Fig. 1b). NK cells were accumulating in BCF already in acute SIVagm infection (day 9 p.i., Supplementary Fig. 1a, b) and this was again not seen in acute SIVmac infection (Supplementary Fig. 1b, c). CXCR5 is a chemokine receptor allowing migration of B cells and T$_{FH}$ into BCF. In line with the spatial dynamics of the NK cells, the proportion of CXCR5+ NK cells among total NK cells were increased after SIVagm infection in mesLN of AGM as compared to non-infected animals (Fig. 1c, Supplementary Fig. 1d).

In a previous study, we had depleted NK cells during chronic SIVagm infection through administration of anti-IL-15[17]. This lead to increases in viremia and strong viral replication in

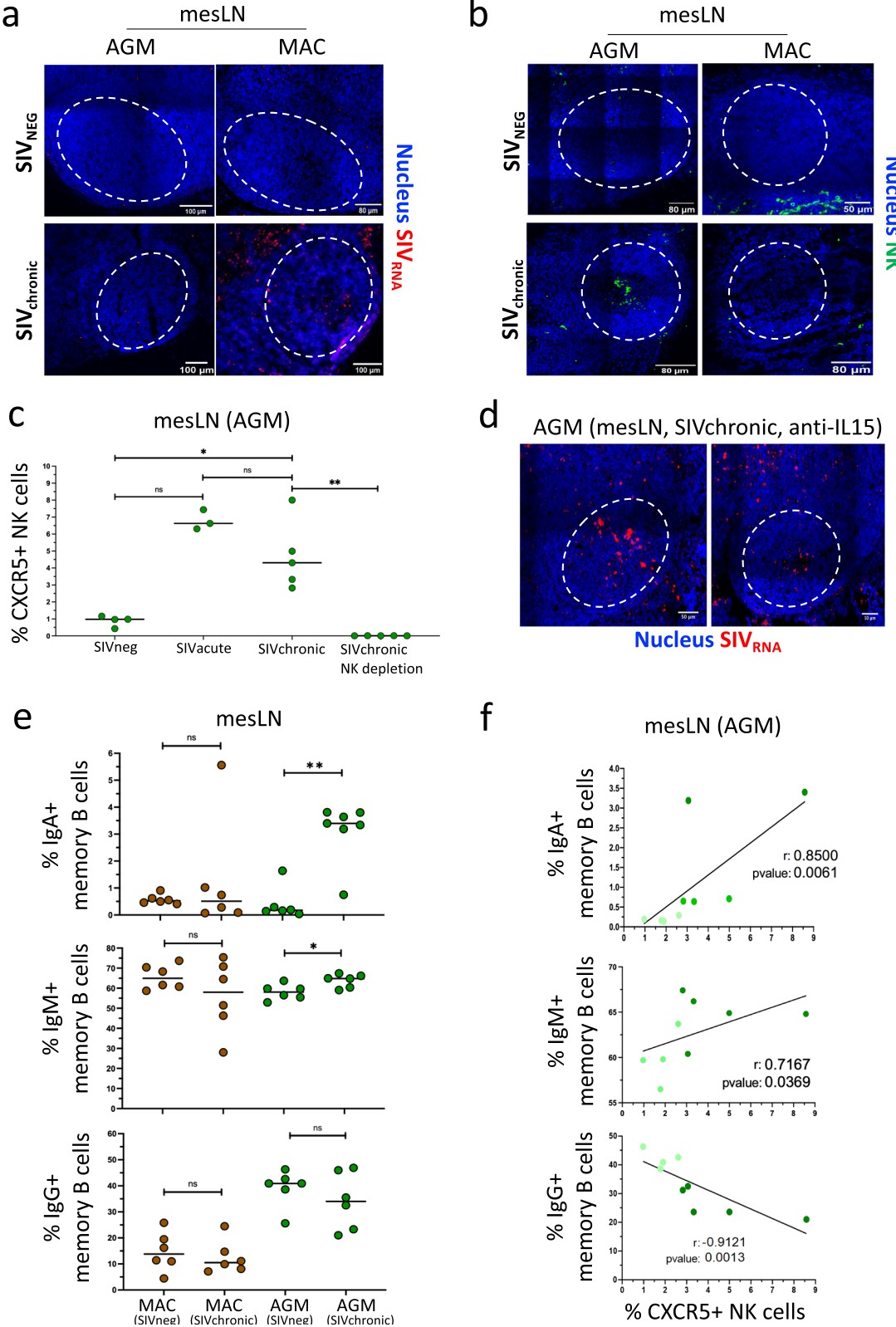

peripheral LN and spleen of AGM[17]. Here, we analyzed in the same, anti-IL-15 treated animals, the expression of SIV RNA in situ in the mesLN and observed signs of strong viral replication in their BCF (Fig. 1c, d). Thus, NK cells controlled SIVagm replication not only in peripheral LN and spleen, as reported previously[17], but also in the LN draining the intestine. Altogether, our results demonstrate that CXCR5+ NK cells were rapidly

induced in mesLN upon SIVagm infection and that the NK cells protected BCF from viral replication in mesLN.

**Memory B cell IgA+ production is increased in mesLN during SIVagm infection**. We raised the question if there was a link between the NK cell-mediated control in mesLN and the humoral responses. We investigated B cells and IgA responses in mesLN

**Fig. 1 SIV replication, NK cells and IgA+ B cells in mesenteric lymph nodes at steady state and during SIVagm and SIVmac infection.**
**a** Immunofluorescence staining of viral RNA in BCF from mesLN of AGM and MAC. Four uninfected (SIVneg) and four chronically infected (SIVchronic) animals per species were studied. On the tissue sections, nucleus is stained in blue, RNASIV in red and dash white circles delineate BCF areas.
**b** Immunofluorescence staining of NK cells in mesLN. Four uninfected (SIVneg) and four chronically infected (SIVchronic) animals per species were studied. Nucleus is in blue; NK cell is in green; and dash white circle delineates the BCF areas. **c** Evaluation of CXCR5+ NK cell frequencies in mesLN in non-infected AGM (SIVneg), at day 9 p.i. (SIV acute), in chronic SIVagm infection (SIVchronic) and in chronically SIVagm-infected AGM depleted for NK cells (SIVchronic NK depletion). Percentages of CXCR5+ NK cells among total NK cells are shown. Each green circle represents one individual AGM ($n = 4$ non-infected animals, $n = 3$ acute animals, $n = 5$ chronic animals and $n = 5$ NK depleted chronic animals). A nonparametric Mann–Whitney test ($p \leq 0.05 = *$; $p \leq 0.01 = **$; $p \leq 0.001 = ***$) was used. **d** Immunofluorescence staining of viral RNA in mesLN from chronically SIVagm-infected AGM depleted for NK cells. The mesLN were collected from four previously reported animals[17]. Nucleus is stained in blue, RNASIV in red. The dash white circle delineates the B cell follicle area. Two representative BCF are shown from 2 distinct animals. **e** Measurement of IgA+, IgM+ and IgG+ memory B cells in mesLN from non-infected (SIVneg) and from chronically infected (SIVchronic) MAC (brown) and AGM (green). Percentages among memory B cells in mesLN are shown. A nonparametric Mann–Whitney test ($p \leq 0.05 = *$; $p \leq 0.01 = **$; $p \leq 0.001 = ***$) was used. Each dot indicates an individual animal ($n = 6$ noninfected per species, and $n = 6$ chronic animals per species). **f** Correlation between CXCR5+ NK cells and IgA+, IgM+ and IgG+ memory B cell in mesLN from non-infected (light green) and chronically SIVagm-infected AGM (dark green). Each green circle indicates an individual animal ($n = 4$ non-infected AGM, and $n = 5$ chronic AGM). Spearman $r$ test was used ($p \leq 0.05 = *$; $p \leq 0.01 = **$; $p \leq 0.001 = ***$). Animals and time points of tissue collections are described in Supplementary Data 1, 2.

during SIVagm and SIVmac infection. Uninfected ($n = 6$ animals per species) and chronically SIV-infected animals (6 AGM, 6 MAC) were analyzed. B cells were defined as CD45+CD3⁻CD20+ cells and memory B cells as CD45+CD3⁻CD20+IgD⁻ cells (Supplementary Fig. 1e). The frequency of total B cells did not vary significantly following SIVagm or SIVmac infection in mesLN or any intestinal compartment analyzed when compared to uninfected animals (Supplementary Fig. 1f, g). In the absence of total numbers, we cannot exclude that some changes were not detected or underestimated. Then, we analyzed the isotype of membrane-bound Ig expressed by memory B cells from mesLN. IgA class switching is known to be altered in intestinal lymphoid tissues of chronic SIVmac and HIV-1 infections[31,36]. We observed no differences between non-infected and SIV-infected MAC regarding IgA+, IgG+ or IgM+ memory B cells in mesLN (Fig. 1e). There was also no change concerning IgG-expressing memory B cells in SIVagm-infected AGM, but an increased frequency of IgM and of IgA expressing memory B cells compared to non-infected AGMs (Fig. 1e). In the absence of total numbers, we cannot exclude that some changes were overestimated. Of note, the frequency of IgA+ and IgM+ memory B cells from mesLN in SIVagm infection correlated positively with the frequency of CXCR5+ NK cells also present in mesLN (Fig. 1f). Moreover, the frequency of IgG+ memory B cells from mesLN in SIVagm infection correlated negatively with the frequency of CXCR5+ NK cells present in mesLN (Fig. 1f). These correlations were seen when we combined all animals (infected and uninfected) and were lost when analyzing only infected animals, except for IgG, for which a trend for negative correlation persisted ($p = 0.07$, $r = 0.87$) (Supplementary Fig. 2a–c). These results showed that the frequencies of IgA+ cells among memory B cells were increased in mesLN following SIVagm infection in contrast to SIVmac infection, where such frequencies remained low and did not increase.

**Loss of total IgA in the intestinal lumen during chronic SIV-mac in contrast to SIVagm infection.** We then explored the total IgA response from intestine in SIV infection. We analyzed in parallel the ileum, jejunum, and colon in non-infected, acutely, and chronically infected animals. The analysis of spatial IgA distribution by immunohistochemistry demonstrated the presence of IgA (positive cells and antibodies) both within the lamina propria and in the lumen of the gut from non-infected AGM and MAC (Supplementary Fig. 3a). During acute infection, no differences regarding the level nor the location of IgA were observed in AGM or MAC when compared to uninfected animals

(Fig. 2a–c, Supplementary Fig. 3a). However, the IgA staining was drastically decreased in chronic SIVmac infection compared to non-infected MAC in the intestinal compartments, unlike to chronic SIVagm-infected animals (Fig. 2a–c, Supplementary Fig. 3a). Only scarce IgA+ cells were still detected in the lamina propria of intestinal tissues from chronic SIVmac-infected animals and the IgA signal was massively lost from the lumen (Fig. 2d). In contrast, normal IgA signals were detected in chronic SIVagm infection in all intestinal compartments and within the gut lumen (Fig. 2a–d, Supplementary Fig. 3a).

We further analyzed IgA expressing cells in the intestine (jejunum, ileum, and colon) by quantifying Ig expressions on memory B cells using flow cytometry (Supplementary Fig. 3b, c). In MAC, the frequency of IgM+ memory B cells remained comparable between non-infected and chronically infected animals (Supplementary Fig. 3b), while IgG+ memory B cells increased in jejunum and colon from MAC SIV-infected compared to uninfected animals (Supplementary Fig. 3b). In contrast, IgA+ memory B cells decreased in colon from SIV-infected MAC compared to uninfected animals (Supplementary Fig. 3b). In AGM, the frequencies of IgG+, IgA+ and IgM+ memory B cells remained comparable between non-infected and chronically infected animals (Supplementary Fig. 3c). Altogether, AGM displayed normal levels of IgA production in all three intestinal compartments analyzed, and showed no increases of IgG after infection, in contrast to MAC. In MAC, IgA levels were maintained in acute infection but decreased in chronic SIVmac infection, and SIgA seemed to be particularly lost in the gut lumen.

**Differences in Ig profiles between blood and intestine-associated tissues during SIVagm and SIVmac infections.** We then investigated whether the alterations in Ig profiles observed in the gut-associated lymphoid tissues were reflected in the blood. We focused on IgA+ and IgG+ memory B cells. Their frequencies remained comparable between uninfected and infected animals in the blood, for both MAC and AGM (Supplementary Fig. 4a–f). We also measured the concentrations of circulating IgG and IgA in plasma from infected AGM and MAC as well as in control animals. There was no difference between non-infected and SIV-infected AGM for plasma IgA and IgG titers (Fig. 3a, Supplementary Fig. 4g). In contrast, there was a decrease of circulating IgA ($p = 0.0065$) in plasma from SIVmac-infected MAC compared to non-infected MAC (Fig. 3a).

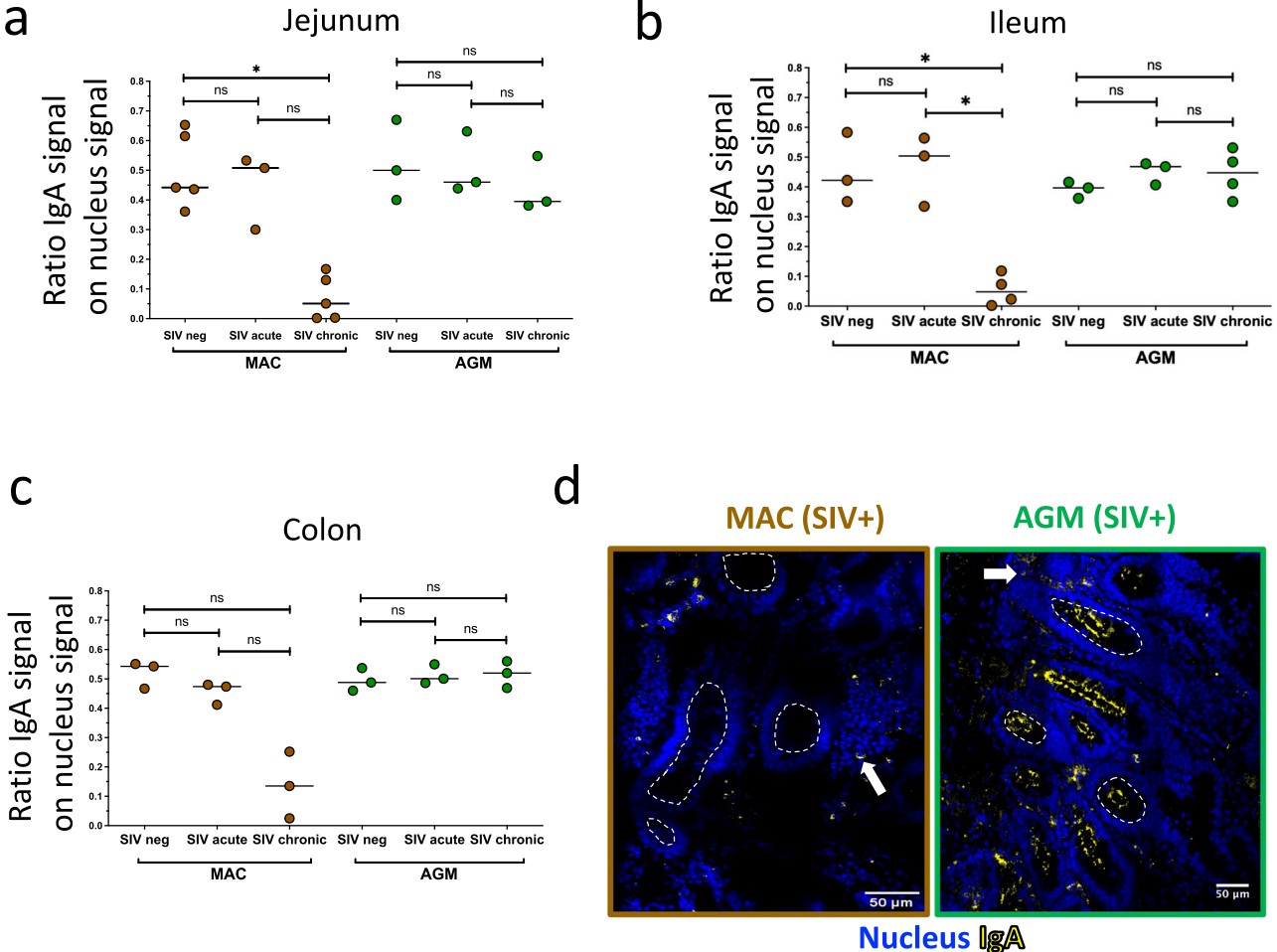

**Fig. 2 Ig dynamics and anatomical distributions during SIVagm and SIVmac infections in the intestine. a–c** Longitudinal evaluation of IgA levels in the intestine. The ratio of the IgA signal per nucleus signal was measured by microscopy in **a** jejunum, **b** ileum, and **c** colon in non-infected (SIVneg) and SIV-infected at day 9 p.i. after infection (SIVacute) and chronic SIV infection (SIVchronic) in MAC (brown) and AGM (green). Three non-infected MAC (ileum, colon), five non-infected MAC (jejunum) and three non-infected AGM (jejunum, ileum, colon) were analyzed. Three MAC (jejunum, ileum, colon) and three AGM (jejunum, ileum, colon) were analyzed at day 9 p.i. Three chronic MAC (colon), four chronic MAC (ileum), five chronic MAC (jejunum), four chronic AGM (ileum) and three chronic AGM (jejunum, colon) were analyzed. A nonparametric Mann–Whitney test ($p \leq 0.05 = *$; $p \leq 0.01 = **$; $p \leq 0.001 = ***$) was used. **d** Enlargements of the immunofluorescence staining of IgA (yellow) and nucleus (blue) in the colon from chronically SIVmac-infected MAC and chronically SIVagm-infected AGM (three chronic MAC and three chronic AGM). Arrows represent IgA+ cells and circles delineate the lumen zone. The images show a representative section in the ileum for one MAC and one AGM.

**Inflammation in SIVmac infection correlated with lower blood IgA levels.** Because IgA are associated with microbiota control and an anti-inflammatory function[21,24], we analyzed if there was an association between IgA levels and inflammation. The concentrations of systemic markers of inflammation and microbial translocation (sCD14, LPS-binding protein (LBP) and intestinal fatty-acid binding protein (I-FABP)) were quantified in the plasma by ELISA (Fig. 3b, Supplementary Fig. 4h, i). These markers are generally used to evaluate inflammation levels associated with monocyte/macrophage activation (sCD14), intestinal epithelial barrier disruption (I-FABP) and microbial translocation (sCD14, I-FABP, LBP)[28,29]. As expected, we detected an increased concentration of plasma sCD14 in chronic SIVmac infection as compared to baseline levels but not in SIVagm infection (Fig. 3b). For the two other markers, LBP and I-FABP, the increases in chronic SIVmac infection were not statistically significant (Supplementary Fig. 4h, i). There could be several reasons for that. It might be that the number of animals studied was too limited to see a statistically significant difference. It could also be that the animals have not yet enough progressed toward

disease, due to the species (cynomolgus macaque) and virus (SIVmac251), which are known to progress generally more slowly to disease than rhesus macaques infected with SIVmac239.

We analyzed if the levels of IgA and IgG correlated with those of the systemic markers analyzed (Fig. 3c, d, Supplementary Figs. 5, 6). Strikingly, plasma IgA and sCD14 levels correlated negatively, and plasma IgG, which can be a marker for hypergammaglobulinemia in HIV/SIVmac infections[37], correlated positively with LBP in MAC (Fig. 3c, d). When analyzing the data of the infected animals only, there was still a strong positive correlation of IgG with LBP ($p = 0.001$, $R = 0.95$), and a trend for positive correlation with sCD14 in the MAC ($p = 0.09$) (Supplementary Fig. 6a). In contrast, there was no correlation between any of these inflammatory markers and Ig levels in SIVagm infection (Supplementary Figs. 5b and 6b). These data support the association between systemic inflammatory levels and abnormal isotypic Ig profiles in SIV infection.

**Stronger SIV-ENV specific memory B cell responses in mesLN and intestine during SIVagm than SIVmac infection.** We then

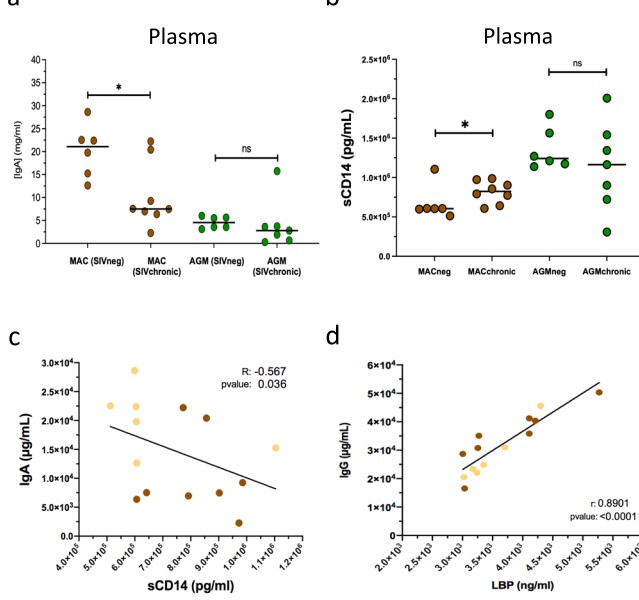

**Fig. 3 IgA titers negatively correlated with sCD14 in peripheral blood during SIVmac infections. a** IgA titers in plasma from MAC and AGM (non-infected (SIVneg) and chronically SIV-infected (SIVchronic)) ($n = 6-8$ animals). A nonparametric Mann–Whitney test ($p \leq 0.05 = $*; $p \leq 0.01 = $**; $p \leq 0.001 = $***) was used. **b** Plasma titers of sCD14 from non-infected (SIVneg) and chronically SIV-infected (SIVchronic) MAC and AGM ($n = 6$ non-infected animals per species, $n = 8$ chronic MAC and $n = 7$ chronic AGM). From Figure A to C, a nonparametric Mann–Whitney test ($p \leq 0.05 = $*; $p \leq 0.01 = $**; $p \leq 0.001 = $***) was used. Each dot represents an individual animal. **c** Correlation between microbial translocation marker sCD14 in plasma and plasmatic-IgA in MAC (non-infected animals in light brown and chronically infected animals in dark brown) ($n = 6-8$ animals). **d** Correlation between inflammatory (LBP) and plasmatic-IgG in MAC (non-infected animals in light brown and chronically infected animals in dark brown) ($n = 6-8$ animals). Spearman $r$ test was used ($p \leq 0.05 = $*; $p \leq 0.01 = $**; $p \leq 0.001 = $***). Each dot represents an individual animal.

measured and compared the SIV-specific Ig levels in the intestine and mesLN of AGM and MAC. Recombinant GP140-foldon Env GP140 proteins from SIVagm and SIVmac were constructed and used for measuring GP140-specific memory B cells by flow cytometry. As expected, the frequencies of GP140-specific memory B cells were increased in blood following SIVagm and SIVmac infections as compared to uninfected animals (Fig. 4a, b). The frequencies of GP140-specific memory B cells were also increased in the mesLN and in all studied intestinal compartments (ileum, jejunum, colon) in SIVagm-infected as compared to uninfected AGM (Fig. 4c–e). In contrast, there were no statistically significant increases in chronic SIVmac infection in mesLN and intestinal mucosa (Fig. 4c, d).

We evaluated the distribution of IgA+, IgM+ and IgG+ GP140-specific memory B cells in chronically-SIV infected animals in blood, mesLN and intestine. IgG+ and IgM+ GP140-specific memory B cells were the most frequent, with IgG+ GP140-specific memory B more predominant in blood, while IgM+ GP140-specific memory B cells were predominant in mesLN and intestinal mucosa (Supplementary Fig. 7a–d). The frequencies of IgA+ GP140-specific memory B cells were very low. In blood, their median levels were 4.2% for MAC and AGM (Supplementary Fig. 7a, b). In the gut-associated-tissues, they were higher than in blood, the median levels ranging between 4.2% and 8.5% (Supplementary Fig. 7c, d). Among all compartments, the frequencies of IgA+ GP140-specific memory B cells

were highest in the jejunum of chronically SIVagm-infected AGM with 8.5% of median (Supplementary Fig. 7d). We quantified secreted GP140-specific Ig in plasma. Both MAC and AGM mounted GP140-specific antibody responses following SIVmac and SIVagm infection, respectively, while the titers in non-infected animals did not increase (Supplementary Fig. 7e, f). As expected, the reactivity of the IgGs was higher than those of IgAs. The ratios of specific versus total Ig (GP140-IgA/IgA and GP140-IgG/IgG) in plasma displayed a similar dynamic between AGM and MAC (Supplementary Fig. 8a).

Finally, we evaluated if there was an association between the plasma GP140-specific Ig profiles and the levels of inflammation or microbial translocation in infected animals (Supplementary Fig. 8b, c). There was a trend for a negative correlation ($r = -0.67$; $p = 0.08$) between anti-GP140 IgA antibodies and plasma I-FABP in SIVmac-infected macaques (Supplementary Fig. 8b).

SIV-specific antibodies in conjunction with NK cells could also mediate ADCC. We analyzed *CD89* gene expression in the NK cells, because IgA driven antibody-dependent NK cell responses have been described to occur via CD89. The NK cells from LN and blood expressed the *CD89* transcript during SIVagm infection (Supplementary Data 5). The *CD89* transcripts were lower than the *NKG2A* transcript, but higher than the *IL-10* transcript, suggesting a possible protein expression of CD89.

## Discussion

Our study demonstrates major differences regarding the IgA+ memory B cells in the mesLN and intestine between non-pathogenic SIVagm and pathogenic SIVmac infection. Previous studies have already analyzed IgA and B cell responses during SIVagm and SIVmac infections in distinct body compartments, including blood, milk, saliva, vaginal and rectal secretions[29,37,38]. These previous reports have shown that SIV-infected AGMs display robust Env-binding antibody responses in milk, higher than those in macaque and humans, which may contribute to the absence of postnatal transmission reported for natural SIV hosts[39]. Here, we investigated whether NK cells have the capacity to migrate into BCF of mesLN and control SIVagm infection. Moreover, we performed a comprehensive analysis of total and ENV-specific IgA and B cell responses in several tissues: blood, gut-associated secondary lymphoid organs (mesLN) and three intestinal compartments (ileum, jejunum, colon). Furthermore, we analyzed if there was an association between NK cells and humoral responses in the mesLN and/or between the IgA responses and systemic inflammation.

Our study uncovered that NK cells migrate into BCF of mesLN during SIVagm infection where they control viral replication, and that they accumulate already in acute infection in mesLN BCF in response to SIVagm infection. We show that IgA+ memory B cells were increased in mesLN during SIVagm infection in contrast to SIVmac infection. Moreover, normal intestinal IgA levels (jejunum, ileum, and colon) were maintained during acute and chronic SIVagm infection, while they strongly decreased in chronically SIVmac-infected macaques compared to healthy macaques. Our data in the pathogenic SIV model are in agreement with previous reports on HIV-1 infection describing that the infection does not induce vigorous specific IgA responses in any body fluid examined[28,31]. Our study underlines that in contrast to pathogenic HIV-1/SIVmac infections in humans and macaques, IgA+ memory B cells in the mesLN are protected during SIVagm infection in AGM. The levels of CXCR5+ NK cells and IgA+ memory B cells in mesLN were correlated, suggesting that the NK-cell mediated viral control participates in the protection and preserved function of the mucosal B cell responses

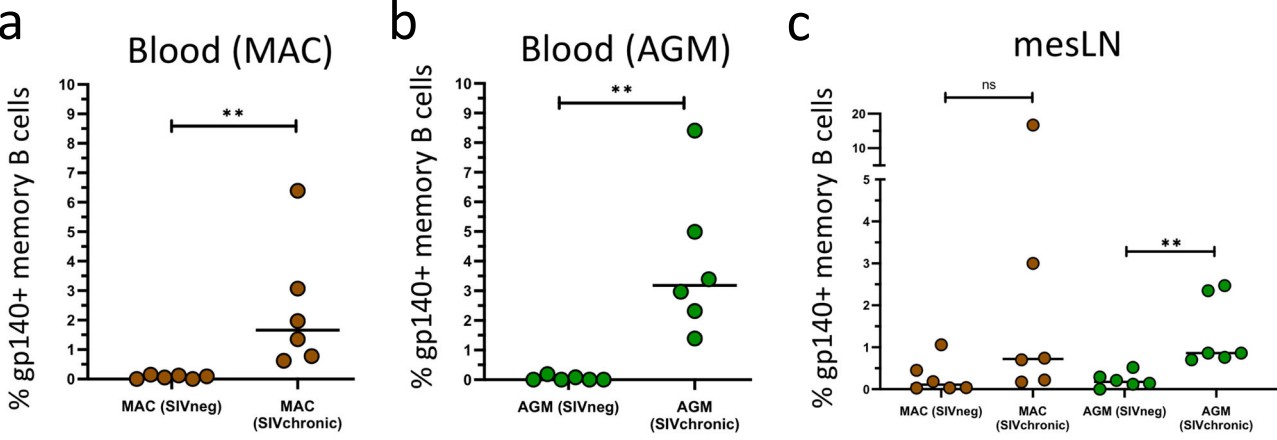

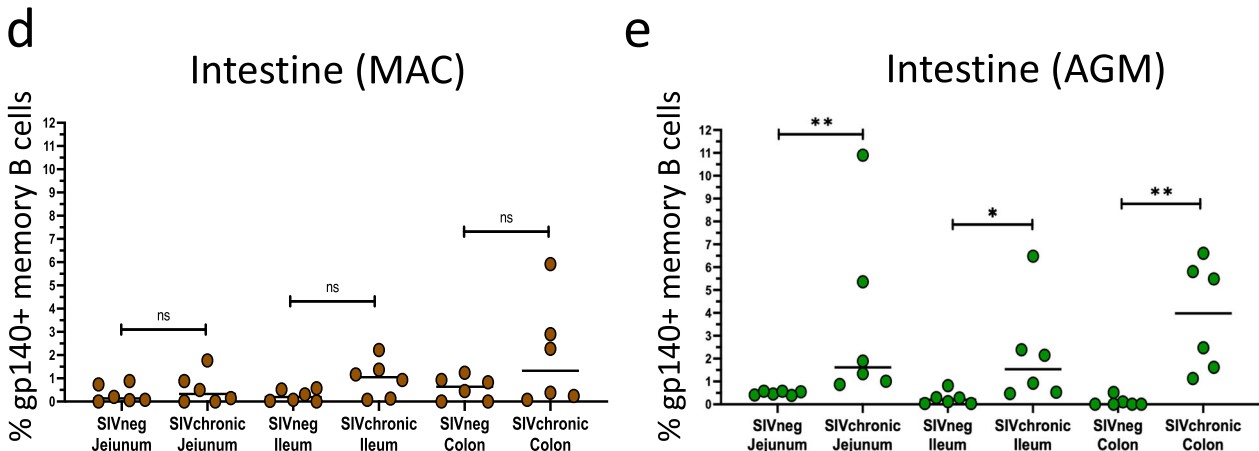

**Fig. 4 SIV ENV-specific memory B cells in blood, mesenteric lymph nodes and intestinal tissues during SIVmac and SIVagm infections. a** Percentage of GP140-specific memory B cells in blood from MAC (non-infected (SIVneg) and chronically infected (SIVchronic)) ($n = 6$ animals). A nonparametric Mann–Whitney test ($p \leq 0.05 = *$; $p \leq 0.01 = **$; $p \leq 0.001 = ***$) was used. **b** Percentage of GP140-specific memory B cells in blood from AGM (non-infected (SIVneg) and chronically infected (SIVchronic)) ($n = 6$ animals). A nonparametric Mann–Whitney test ($p \leq 0.05 = *$; $p \leq 0.01 = **$; $p \leq 0.001 = ***$) was used. **c** Percentage of GP140-specific memory B cells in mesLN from AGM (non-infected (SIVneg) and chronically infected (SIVchronic)) and MAC (non-infected (SIVneg) and chronically infected (SIVchronic)) ($n = 6$ animals per species). A nonparametric Mann–Whitney test ($p \leq 0.05 = *$; $p \leq 0.01 = **$; $p \leq 0.001 = ***$) was used. **d** Percentage of GP140-specific B cells in the intestine from MAC (non-infected (SIVneg) and chronically infected (SIVchronic)) ($n = 6$ animals). A nonparametric Mann–Whitney test ($p \leq 0.05 = *$; $p \leq 0.01 = **$; $p \leq 0.001 = ***$) was used. **e** Percentage of GP140-specific B cells in the intestine from AGM (non-infected (SIVneg) and chronically infected (SIVchronic)) ($n = 6$ animals). A nonparametric Mann–Whitney test ($p \leq 0.05 = *$; $p \leq 0.01 = **$; $p \leq 0.001 = ***$) was used.

in non-pathogenic SIV infection. However, whether there is a causative link with NK cell' capacity to migrate into BCF and/or an indirect link with NK cell-mediated viral control remains to be investigated in the future. The strong control of viral replication in BCF during chronic SIVagm infection as opposed to untreated HIV-1/SIVmac infections indeed could indirectly lead to less chronic inflammation and avoid destruction of the anatomical architecture, in particular through fibrosis and disruption of the FDC network in AGMs, in contrast to the LN damages initiated already during early HIV infection[40], thereby also protecting B cells in BCF in AGMs during SIVagm infection[37]. The natural hosts thus seem to display an original resilience mechanism. They develop a tissue-specific viral control limited to secondary lymphoid organs. This would lead to preservation of normal antiviral adaptive immune responses, including the IgA responses in the gut. These IgA responses could then play a key role in the

control of bacterial translocation and inflammation, which then participates in the maintenance of an intact epithelial barrier despite continuous replication of SIVagm. The continuous SIVagm replication in the gut allows the virus to achieve sufficiently high levels in blood to be more transmissible. Altogether, our data suggest a tissue-specific viral control with distal, indirect impact on other tissues. Our results do not exclude other non-mutually exclusive or synergistic mechanisms of protection of the intestinal tissue, such as strong repair mechanisms in AGM[41], protection of Th17 cells, less IFN-γ responses and regulatory NKG2A+ CD8+ T cells[17,19,42–44].

Moreover, it is not excluded that NK cells also directly modulate B cell responses, including in HIV infection[45]. It has been shown that NK cells can influence humoral responses by diminishing $T_{FH}$ cells and that they can also limit germinal center reaction[46,47]. Also, a crosstalk between NK and B cell has

been described that could impact isotype class switch in B cells[43,48,49]. Aberrant class switching from IgM to IgG rather than to IgA is a hallmark of chronic inflammatory diseases in the gut[50]. The induction of CXCR5+ NK cells capable of migrating into BCF of mesLN might facilitate the impact of NK cells on B cell responses.

It has been reported that depletion of B cells in SIVagm infection leads to moderately increased viral production in the intestine[51,52]. Thus, a partial, although weak contribution of B cells to viral replication in the gut during SIVagm infection is not excluded and could indirectly participate to inflammatory control. However, even if AGM develop better GP120 responses, they do not seem to develop more or better circulating neutralizing antibodies than PLWH[53].

We cannot exclude a stronger IgA driven antibody-dependent NK cell response in SIVagm compared to SIVmac infection. We reported that SIVagm infection in AGM is associated with induction of NK cells in LN that express more frequently Fc-gamma receptors (CD16, CD32, CD64)[17,54]. These highly differentiated NK cells in the AGM LN also expressed a transcriptome profile that is reminiscent of NK cells with ADDC activity (decreased FcεR1 and increased CD3e expression[54]). In general, it is known that mature NK cells display indeed an increased ADCC activity. Future studies on the IgA and IgG dependent ADCC activities of NK cells in these animal models are warranted.

Of note, our data can provide an additional explanation of why early ART has a benefit regarding treatment initiated in chronic HIV infection. Since early ART is known to protect BCF structures in the intestine[40], thereby most likely improving IgA responses, early ART might increase through IgA the control of chronic inflammation and also Env-specific antibody responses in the gut-associated tissues[30].

Blood did not reflect well the IgA+ memory B cell profiles in the intestine. More than 90% of gut Igs are produced in situ by plasma cells in contrast to genital mucosa, where half of the Igs are coming from the blood[55]. This could explain why B cells in blood did not accurately reflect the dynamics in the intestine and further demonstrate the importance to analyze intestinal tissue for understanding tissue-specific physio-pathological virus-host interactions.

The decrease of IgA signal in lumen area from chronically SIVmac-infected macaques was observed in all three intestinal compartments studied (ileum, jejunum, colon). The results we observed in SIVmac infection resemble those previously described in HIV infection[28,29]. The decrease of the IgA production was predominantly due to a strong loss in the intestinal lumen, and not in the lamina propria, even if IgA+ memory B cells seemed to also decrease in SIV chronically infected macaques compared to healthy macaques, at least in the colon. A major difference between non-pathogenic and pathogenic SIV infection was thus the selective maintenance of IgA in lumen during SIVagm infection. The loss of IgA in lumen in SIVmac infection might be due to decreased levels of plasma cells, the major producers of Ig. Plasma cells were not analyzed here, as also often not by others in previous studies in monkeys because of a lack of anti-CD19 antibodies specific for plasma cells from these non-human primates. We recently analyzed though CD138+ cells (another marker of plasma cells) in peripheral LN during SIVagm and SIVmac infection[43]. IL-6 producing CD138+ cells were more frequent in SIVagm than SIVmac infection in the pLN, further suggesting that the loss of IgA in lumen in SIVmac infection could eventually be related to reduced levels of plasma cells, while these are maintained in SIVagm infection. It will thus be interesting in the future to investigate more closely the role of plasma cells in the regulation of inflammation within the intestine and to better understand why their levels are maintained in SIVagm infection.

Total IgA showed stronger negative correlations with sCD14 and LBP in SIVmac infection, than ENV-specific IgA. Also, the ratios of total IgA/IgG were decreased in SIVmac infection, but not that of GP140-specific IgA/IgG. This suggests that the inflammatory control in SIV infection in the intestine is more associated with the maintenance of total IgA responses than with SIV-specific IgA responses. The total, non-SIV-specific IgA responses in the intestine during SIVagm infection might strongly contribute to protect against translocation of bacteria together with the other proposed mechanisms protecting the epithelial barrier.

A limitation of our study is that we did not analyze Peyer's patches. The latter have been known for being major sites for induction of IgA+ memory B cells[56–58]. The tissue sections we analyzed from the AGMs did not contain Peyer's patches and such an analysis was not feasible. However, B cells are known to rapidly migrate from Peyer's patches to the regional mesLN[58], which we studied here. A recent study moreover revisited the role of mesLN and showed that mesLN and not Peyer's Patches were the major inductive site of anti-rotavirus IgA responses[59].

Overall, we show that IgA+ memory B cells were normal or increased in the mesLN and intestine during SIVagm infection. AGMs may have developed resilience mechanisms where tissues are synergistically protected. Thus, NK cells control viral replication in LN, including mesLN as shown here. This could indirectly protect immune responses, such as the SIgA production in the intestine, contributing to the maintenance of bacterial control and epithelial barrier integrity, despite viral replication in the intestinal mucosa. In contrast, mesLN are major viral reservoirs in pathogenic HIV/SIV infections[60] and IgA responses in intestine are damaged[28].

In conclusion, our study suggests an original distal NK-cell mediated mechanism to protect IgA responses. We uncovered indeed that NK cells of mesLN upregulate CXCR5 in SIVagm infection and rapidly migrate into mesLN BCF. This study lays the groundwork for future studies exploring direct or indirect mechanisms on potential consequences for the protection of intestinal IgA responses and inflammation control and on NK-B cell interactions. The development of strategies for enhancing HIV-specific IgA response in mucosa has been largely discussed in the context of vaccine studies[28]. Here our data suggest that total non-specific IgA contributes to the inflammatory control. Strategies allowing early and persisting viral control in BCF and/or restoring normal total IgA levels in the intestine could reveal beneficial for reducing inflammation in virologically controlled PLWH with persistent residual inflammation and in the research for therapies toward HIV cure.

## Methods

**Monkeys, SIV infection and ethics statement.** Twenty-seven AGMs (Caribbean *Chlorocebus sabaeus*) were included in this study. Seven AGM were used as uninfected controls and twenty AGM were infected with SIVagm. Twenty-seven cynomolgus MACs (*Macaca fascicularis* [MAC]) imported from Mauritius island were included in the study, out of which twelve MAC were used as uninfected controls and fifteen MACs were infected with SIVmac. Macaques with a controller genotype (H6) were not included in the study. The AGMs were infected with the SIVagm.sab92018 wildtype isolate and the MAC with the SIVmac251 isolate as previously reported[17]. The description of the animals is shown in Supplementary Data 1, 2. The other AGMs treated with anti-IL15 monoclonal antibody derived from previous studies presenting a deep NK cell depletion[17]. The five NK cell-depleted AGMs were chronically infected with SIVagm.sab92018 for 1–3 years at the time of anti-IL15 administration[17].

The AGM and MAC were housed in IDMIT infrastructure facilities (CEA, Fontenay-aux-Roses, France) under animal facility authorization #D92-032-02 (Prefecture des Hauts de Seine, France) and in compliance with European Directive 2010/63/EU, the French regulations and the Standards for Human Care and Use of Laboratory Animals, of the Office for Laboratory Animal Welfare (OLAW, assurance number #A5826-01, US). The study was approved by the institutional

ethical committee "Comité d'Ethique en Expérimentation Animale du Commissariat à l'Energie Atomique et aux Energies Alternatives" (CEtEA #44). Monitoring of the monkeys was under the supervision of the veterinarians in charge of the animal facilities. Animal experimental protocols were approved by the Ethical Committee of Animal Experimentation (CETEA-DSV, IDF, France; Notification 12-098). The samples from the animals included here were from animals purchased and used for other studies. These studies were approved and accredited under statement numbers A16-016, A17-059, A12-006, A17-044, A13-005 and A15-035 by the ethics committee, registered and authorized under Number 44 at the French Ministry of Education and Research with the reference numbers APAFIS#2453-2015102713323361, APAFIS#4442-2016030818243239 and APAFIS# 11236-2017091214402801 and APAFIS#319-2015031314518254.02. Some results of these other studies are already published[42,61–63]. Animals were handled by veterinarians in accordance with national regulations (CEA Permit Number A 92-32-02) and the European Directive (2010/63, recommendation no. 9) and in compliance with the Standards for Human Care and Use of Laboratory of the Office for Laboratory Animal Welfare (OLAW, USA) under OLAW Assurance number #A5826-01. Animals were housed in ASL3 confinement in adjoining individual cages allowing social interactions, and maintained under controlled conditions with respect to humidity, temperature, and light (12 h light/12 h dark cycles). Water was available ad libitum. Animals were monitored and fed once or twice daily commercial monkey chow and fruit by trained personnel. Environmental enrichment was provided including toys, novel foodstuffs, and music under the supervision of the CEA Animal Welfare Body. Experimental procedures (animal handling, viral inoculations, and samplings) were conducted after sedation with ketamine chlorhydrate (Rhone-Merieux, Lyon, France, 10 mg/kg). Tissues were collected at necropsy: animals were sedated with ketamine chlorhydrate 10 mg/kg) then humanely euthanized by intravenous injection of 180 mg/kg sodium pentobarbital. The time of necropsy is indicated in Supplementary Data 1, 2. The sample size varied between three and nine monkeys per group (n = 6 animals in most experiments). Sample analyses were performed in random distributions into groups and random order, according to the tripartite harmonized International Council for Harmonization of Technical Requirements for Pharmaceuticals for Human Use (ICH) Guideline on Methodology (previously coded Q2B). The investigators were not blinded while the animal handlers were blinded to group allocation.

**Tissue collections and processing**. Plasma was obtained by blood centrifugation 1800 x g for 10 min and stored at −80 °C. PBMCs were isolated on density gradient (Eurobio). The cells were conserved in fetal bovine serum (FBS) (Eurobio) with 10% of DMSO (Sigma) in liquid nitrogren. The tissue samples (mesLN, Jejunum, Ileum, and Colon) were cut and grinded with the help of a gentleMACS Dissociator (Miltenyi Biotec). The cells were then filtered (70 µm, Clearline) and stored in FBS with 10% of DMSO in liquid nitrogen.

To conserve tissues frozen in O.C.T. (Tissue-Tek), pieces were incubated in Paraformaldehyde (PFA) 4% (AlfaAesar) overnight. The next day, they were washed 3 times in PBS with 20 minutes of incubation at room temperature. After that, they were immersed into increasing concentrations of sucrose (Sigma), i.e., 10%, 20%, and 30% at 4 °C. After these steps, the tissues were put into O.C.T. and frozen in ethanol/dry ice solution and conserved at −80 °C.

**Flow cytometry staining**. For the NK and B cell panels, we saturated non-specific sites by monkey sera. Memory B cell population gating in AGM was already realized[33]. For the panels (Supplementary Data 3), we saturated non-specific sites by FBS. Then we added the antibody mixes to the samples (1 million of cells). The B cell panels also included biotinylated GP140 proteins from SIVmac and SIVagm (see below) and streptavidin-PE (ThermoFisher). Cells were permeabilized using Cytofix solution (BD). The fluorescence staining was observed and captured by a flow cytometer (BD, LSR II) and DIVA software (BD). The signals recorded have been analyzed by FlowJo.

**Immunofluorescent staining**. All tissues in O.C.T were cut at 10 µm on a cryostat (LEICA CM 3050 S) and put on slides (Thermo Scientific). These slides were conserved at −20 °C. The epitope revelation protocol was realized with PBS incubation at room temperature for 15 min. Then, we let the slide in methanol (Fisher Chemical) at −20 °C for 2 h and incubated in formaldehyde 2% at room temperature (Sigma) for 15 min. We saturated non-specific sites by BSA 4% (Sigma) during 2 h at room temperature and washed for 1 h with PBS. The primary antibodies (Supplementary Data 4; 1:200) were added overnight at 4 °C. Then we washed the slide during 1 hour in PBS and we added secondary antibodies (Supplementary Data 4; 1:200) and DAPI (Supplementary Data 4; 1:1000) for 1 h at 4 °C. We washed the slide during 1 h in PBS and we added mounting medium (Invitrogen). We finally dropped off the cover (Fisher Scientific, Dutscher) from the slide with mounting medium. The fluorescence staining was observed and captured by a spinning-disk on three tissue sections per monkey (Yokagawa, CellVoyager CV1000). We analyzed these images by ImageJ (Fiji).

**Relative quantification by microscopy staining**. On our acquired samples by ImageJ software (Fiji) the noise was deleted via threshold establishment. The threshold was determined by background staining analysis. Then the median of

intensity fluorescence adjusted without background (MFIadj) was measured on our samples in channels corresponding to DAPI and the given marker to be measured, respectively. Finally, we calculated MFIadjIgA/MFIadjDAPI ratio on each sample.

**Fluorescent In Situ Hybridization**. All tissues in O.C.T were cut at 10 µm on cryostat (LEICA CM 3050 S) and put on slides (Thermo Scientific). These slides were conserved at −20 °C. The epitope revelation protocol was realized by two methods. For detection of SIVmac RNA, we followed the RNAscope protocol with SIVmac251-gag probe utilization (Advanced Cell Diagnostics Europe) and ACD HybEZ Hybridization system (Advanced Cell Diagnostics Europe, 310013). For detection of SIVagm RNA, the SIVagm probe was made based on the SIVagm.-sab92018 backbone, as described in a previous study[17]. The primary antibodies (Supplementary Data 4; 1:200) were added overnight at 4 °C. Then we washed the slide during 1 h in PBS and added secondary antibodies (Supplementary Data 4; 1:200) and DAPI (Supplementary Data 4; 1:1000) for 1 h at 4 °C. We washed the slide during 1 h in PBS and we added mounting medium (Invitrogen, 00-4958-02). We finally dropped off the cover (Fisher Scientific, Dutscher) from the slide with mounting medium. The fluorescence staining was observed and captured by a spinning-disk on three tissue sections per monkey (Yokagawa, CellVoyager CV1000). We analyzed these images by ImageJ (Fiji).

**Plasma IgA and IgG purification**. Plasma IgA and IgG from AGMs and Macaques were purified by batch/gravity-flow affinity chromatography using peptide M-coupled agarose (Invivogen, SanDiego, CA) and protein G Sepharose 4 fast flow beads (GE Healthcare, Chicago, IL) for IgAs and IgGs, respectively.

**Trimeric SIVagm and SIVmac251 GP140 proteins**. To produce g140 foldon-type trimers of SIVmac251 (GenBank# AJP75601.1) and SIVagm (agm.Sab92018, GenBank# ADO34206.1) Env glycoproteins, corresponding codon-optimized DNA fragments designed based on the original construct coding for uncleaved YU-2 trimers[64] were synthesized (Genscript), and cloned into pcDNA™3.1/Zeo(+) expression vector (Thermo Fisher Scientific). Trimeric SIV GP140 proteins were produced by transient transfection of FreeStyle™ 293-F cells using the PEI method[65], and purified by high-performance chromatography using the Ni Sepharose® Excel Resin according to manufacturer's instructions (GE Healthcare). Proteins were controlled for purity by SDS-PAGE and NativePAGE gel staining as previously reported[65], and then biotinylated using BirA biotin-protein ligase bulk reaction kit (Avidity, LLC). Biotinylated SIV GP140 trimers were dialyzed against PBS using Slide-A-Lyzer® Cassettes (35 K MWCO, Thermo Fisher Scientific), and final protein concentrations were measured using a NanoDrop 2000 instrument (Thermo Fisher Scientific).

**Quantification of soluble proteins by ELISA**. The soluble markers of inflammation and microbial translocation in the plasma were measured by ELISA using the following commercial kits: I-FABP (MyBiosource), sCD14 (R&D System) and LBP (Hycult Biotech). They have already been shown to cross-react with AGM samples[66,67].

For quantification of circulating plasma IgA and IgG from AGMs and macaques, high-binding 96-well ELISA plates (Costar, Corning) were coated overnight with 250 ng/well of purified trimeric streptococcal IgA-binding protein (tSAP[68]) or purified goat anti-human IgG antibody (Immunology Jackson ImmunoReseach, 0.8 mg/ml final) for total IgA or IgG titrations, respectively. After washings with 0.05% Tween 20-PBS (Washing buffer), plates were blocked 2 h with 2% BSA, 1 mM EDTA, 0.05% Tween 20-PBS (Blocking buffer), washed, and incubated for 2 h with diluted plasma (1:100 and 1:5000 in PBS for IgA and IgG titrations, respectively) and seven consecutive 1:3 dilutions in PBS. Purified plasma IgA or IgG from AGM and Mac starting at 12 µg/ml and seven consecutive 1:3 dilutions in PBS were used to perform the standard ELISA range. For antibody bindings to SIVagm or SIVmac envelope antigens, high-binding 96-well ELISA plates (Costar, Corning) were coated overnight with 125 ng/well of purified SIVagm/mac envelope proteins. After washings with 0.05% Tween 20-PBS (Washing buffer), plates were blocked 2 h with 2% BSA, 1 mM EDTA, 0.05% Tween 20-PBS (Blocking buffer), washed, and incubated for 2 h with serially diluted purified plasma IgG/IgA antibodies starting at 50 µg/ml and seven consecutive 1:3 dilutions in PBS. After washings, plates were incubated for 1 h with goat HRP-conjugated anti-human IgG or IgA (Immunology Jackson ImmunoReseach, 0.8 mg/ml final) in Blocking buffer, washed, and revealed with HRP chromogenic substrate (ABTS solution, Euromedex). All ELISA experiments were done in duplicates at room temperature using HydroSpeed microplate washer and Sunrise microplate absorbance reader (Tecan Männedorf).

For the quantification of the titers of SIVagm and SIVmac GP140-IgA and GP140-IgG, we constructed SIVagm and SIVmac GP140-foldon Env GP140 proteins using methods as described[69]. For SIVagm, we used as backbone the SIVagm.sab92018 molecular clone[12,70]. GP140 proteins was added and incubated on night at room temperature in plates covered by parafilm (Costar). Then, the same protocol was used like just above for IgA and IgG titer.

**Bioinformatic analysis of the RNA sequence data**. Bioinformatic analyses for evaluation of *CD89* expression were performed using the genome-wide

transcriptomic data deposited by ref. [54] Data were analyzed using R version 3.4.3 and the Bioconductor package DESeq2 version 1.18.1[71]. Normalization and dispersion estimation were performed with DESeq2, using the default parameters, and statistical tests for differential expression were performed by applying the independent filtering algorithm. A generalized linear model, including the monkey identifier as a blocking factor, was used to test for the differential expression between the biological conditions. For each pairwise comparison, raw p Values were adjusted for multiple testing according to the Benjamini and Hochberg (BH) procedure[72]. Each list was used to query the Kyoto Encyclopedia of genes and Genomes (KEGG), GO-biological function database and Wiki pathways. Genes with an adjusted p Value < 0.05 were considered differentially expressed (Supplementary Data 5).

**Statistics and reproducibility**. Comparisons among independent groups were conducted using nonparametric Wilcoxon-Mann-Whitney test ($p \leq 0.05 = *$; $p \leq 0.01 = **$; $p \leq 0.001 = ***$). The p values shown were not corrected for multiple comparisons. The correlation analyses were performed according to Spearman r test. All statistical computations were performed using Prism (GraphPad, La Jolla, CA). The sample size is described in Supplementary Data 1, 2.

**Reporting summary**. Further information on research design is available in the Nature Research Reporting Summary linked to this article.

## Data availability

Source data are provided with this paper (Supplementary Data 6). RNA sequencing raw data has been deposited in the Gene Expression Omnibus database by Huot et al. Nat. Med. 2021 according to the GSE140600 accession number[54].

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

## Acknowledgements

We are grateful for the excellent contributions from the veterinarians and staff at the IDMIT Center (Benoit Delache, Jean-Marie Helies, and Raphaël Ho Tsong Fang). NH was supported by the Fondation J. Beytout and Institut Pasteur. PR was recipient of a PhD fellowship from the University Paris Diderot, Sorbonne Paris Cité and supported by the NIH (R01AI143457) and Institut Pasteur. CaP was recipient of a Roux-Cantarini Postdoctoral Fellowship. Cy.P. was the recipient of an ANRS post-doctoral fellowship. We would like to acknowledge grant support from Sidaction and ANRS to MMT. H.M. received core grants from the Institut Pasteur, the INSERM and the *Milieu Intérieur* Program (ANR-10-LABX-69-01) and was supported by an ANRS grant. We gratefully acknowledge the support to IDMIT from the French government: Investments for the Future program for infrastructures (PIA) through the ANR-11-INBS-0008 grant as well as from the PIA grant ANR-10-EQPX-02-01 to the FlowCyTech facility at IDMIT. We equally acknowledge the Investments for Future grant ANR-10–INSB–04 to support the UtechS Photonic BioImaging (Imagopole) and C2RT facilities at Institut Pasteur.

## Author contributions

P.R., N.H., CyP, H.M., and M.M.T. designed the experiments. P.R., CyP, N.H., B.J., M.L., and CaP performed the experiments. V.C. and B.J. coordinated the animal studies. P.R., CyP, N.H., B.J., A.S.C., H.M., and M.M.T. analyzed the data. B.J., A.S.C., and M.M.T. obtained the funding. P.R. and M.M.T. wrote the manuscript and all co-authors reviewed it.

## Competing interests

The authors declare no competing interests.
