## [Peer Review File · Communications Biology]

Reviewers' comments:

Reviewer #1 (Remarks to the Author):

This is a strong study that examines the tissue interplay between, IgA, NK cells and inflammation after SIV AGM or MAC infection. This is a great example of studying permissive versus restrictive infections in an animal model using more than blood to understand biology and immunology. For the most part, the results are sound and on good scientific rationale and experimentation. The results of these studies could provide a new mechanism for exploration of gut dysbiosis during HIV infection. There are several major revisions that could be made to strengthen the manuscript as well as minor editing in the text and figures for mistakes and clarity.

1. Statistical analysis, a nonparametric rank test such as Wilcoxon-Mann-Whitney should be used for comparisons with $n=4$ or $n=6$. Ideally, correction for multiple comparisons should be done, but due to the exploratory nature of this science this should be explicitly said that these are uncorrected p values. This reanalysis may simply the conclusions and improve the conclusions of the manuscript.
2. Figures need to be revised to have error bars removed and dot plots. Labeling also needs to be clearer and consistent.
3. More details comparing the total number of NK cells or B cells as well as the frequency should be provided and discussed.
4. The study is important but descriptive, claims of causation should be removed. For example, while you observe higher levels of CXCR5 NK cells, there are no studies show a direct regulation in the manuscript. So claims of this regulation or causation of between inflammation should be tempered. This study would lay the groundwork for future studies exploring this direct mechanism.

Below are some minor or specific points that relate to the major ones above.

For the dot plots, would be easier to see if the circles are enlarged and the error bars are removed. I would revise all graphs to be dot plots and not shift with bar or box and whisker.

For statistical tests, I believe a nonparametric test should be used for rank such as the Wilcoxon for this number of animals. This would include the analysis of the supplementary material. More details about the viral infections (route of delivery etc. as well as viral loads from the blood should be shown in the results.

Figure 1 B should be explained in the results. Similarly so should figure 1C. I now see 1C is after 1D, but unclear what this adds to the manuscript. Also the figure is not labeled well.

For figure 1D, what is acute and pos? This is not explained. Also, the total number of NK cells identified should be shown here in addition to the percentage. If the SIV negative do not have many NK cells that could explain the frequency difference.

The last line of the first section of results should be revised, the authors did not show direct evidence for protection of the BCF from viral replication.

Again for figure 1E, the total number of B cells would be helpful to confirm if this was a real observation or simply because of the really low levels of IgA B cells.

The observation of increased IgM should also be included if remain significant because of a new Wilcoxon test. Why only the IgA is the focus?

Figure 2A-C should be dot plots with error bars removed. Labeling should be revised and consistent with previous figures.

Figure 2E-F (why only MAC shown here) Compare with AGM. With revised statistics, this may not be as striking and could be moved to the SOM.

For figure 3 in the blood, why are the uninfected controls included. This will be important since the spread of IgA memory cells seems large.

Again figure labeling needs to be revised for 3G-H. and revised statistics.

At the end of the figure 3 results paragraph the sentence was cut off. Starting with Although,.... GP or gp140 should be used.

Figure 4F/G should be removed since not significant. Could be presented in the SOM.

Reviewer #2 (Remarks to the Author):

Rasclé and colleagues expand on their previous work that showed NK cell mediated control of SIV in peripheral lymph nodes. Here they investigated if the same happened in Mesenteric LN. They also investigated if SIV control help preserved mucosal B cell responses with a focus on IgA and prevent an increase in microbial translocation.

Major comments:

Figure 1 A, B, C: can the results be quantified?

Figure 1F, Figure 3 I and J, and Figure S4 it is not appropriate to group SIV infected and uninfected animal for this type of analysis.

What is the effect of depleting IL-15 on other cell types? Are HIV-specific CD8 T cells also impaired? How specifically is it affecting NK cells?

Sup Figure 3. Levels of sCD14 are high at baseline in AGM, even higher than in SIVmac. How do the authors explain this result?

Unclear what is supporting the statement on the higher affinity of IgG compared to IgA.

The authors suggest that total IgA responses are protecting from microbial translocation but there is no increase in LBP or iFABP in SIVmac to support this concept.

Minor comments:

Introduction: the work from Shulzhenko et al (Nat Med 2011, PMID 22101768) is also worth putting into context.

Figure 1 A and B the color coding in the legend is wrong.

How representative are the Figure 1 A, B, C, and Figure 2 D? There is no indication of how many tissue sections were analyzed. The same applies to the supplementary figures with microscopy images.

There is a sentence missing at the end of the 4th paragraph of the results.

Figure 4 F and G, the N on the panels does not match with what is indicated in the legend. In general, the N for each panel could be better indicated in the legends.

Page 16, flow cytometry does not generate images.

Reviewer #3 (Remarks to the Author):

The authors have provided an interesting report detailing differences between pathogenic and non-pathogenic SIV infections in terms of NK cell frequency and IgA responses in gut associated

lymphoid tissues. My only major concern about the manuscript is that the authors have tried to link the presence of NK cells in the mesenteric lymph node to the presence of IgA responses. I can see nothing in the manuscript other than a correlation that suggests these two factors are linked. The observations could simply reflect reduced damage to the immune system in animals infected with non-pathogenic viruses. As such, I think language about the association should be amended to reflect that the causative nature of the association is unknown.

The presented data raises several questions, which if answered could improve the manuscript.

1 - In Figure 1C the authors show an increase in cells carrying viral RNA in animals depleted of NK cells by anti-IL15 treatment. Do the authors know what happens if anti-IL15 treatment is stopped? Do the viral RNA carrying cells go away - are they eliminated or do they transition to latently infected cells. This could be assessed by comparing DNA levels at different time points. It would be interesting to know if the authors are observing a role for NK cells in eliminating virus infected cells or if they are observing a pro-latency effect, as previously reported for CD8+ T cells in SIV-infected macaques by the laboratory of Gudino Silvestri.

2 - The authors show the presence of NK cells in the mesenteric lymph node associates with reduced cells carrying viral RNA. Is there also a reduced number of cells carrying viral DNA?

3 - Could the observation of higher NK cell numbers in the LN of AGM compared to MAC reflect a species difference upon any infection? What is seen if you infect the two species with a virus other than SIV?

4 - What is known about Fc alpha receptor expression on NK cells in AGM? Could increased frequency of IgA drive antibody-dependent NK cell responses?

Minor comment:

The first paragraph on page 7 appears to be truncated.

We sincerely thank the reviewers for their insightful comments which helped us to improve the manuscript. The changes made in the revised manuscript are indicate in yellow.

Reviewer #1 (Remarks to the Author):

Major comments:

1. Statistical analysis, a nonparametric rank test such as Wilcoxon-Mann-Whitney should be used for comparisons with $n=4$ or $n=6$. Ideally, correction for multiple comparisons should be done, but due to the exploratory nature of this science this should be explicitly said that these are uncorrected p values. This reanalysis may simply the conclusions and improve the conclusions of the manuscript.

The reviewer is right, one should use a nonparametric test. This is indeed what we did, we used the nonparametric Mann-Whitney test (also called Wilcoxon-Mann-Whitney). We are sorry that that was not clear. Given the low numbers in this exploratory study and generally only one parameter compared, correction for multiple comparisons is not recommended. As suggested by the reviewer, we now indicate that all values are uncorrected p values (method section of the revised manuscript).

2. Figures need to be revised to have error bars removed and dot plots. Labeling also needs to be clearer and consistent.

We thank the reviewer for this comment. We changed all the figures according to the reviewer's advices.

3. More details comparing the total number of NK cells or B cells as well as the frequency should be provided and discussed.

Several Figures in the revised manuscript (Fig. 1C and E, Fig.2 E and F, Fig.3A-F, Fig.4A-D and Suppl. Figures 1F and G) show frequencies of NK cells and B cells. We focused here on tissues (and not on blood). We did not include truecount in the flow cytometry analyses and thus unfortunately do not have the total numbers of the cells for the tissues. We mention this limitation now in the revised manuscript (pages 5 and 6: "In the absence of total numbers, we cannot exclude that some changes were not detected or under/over-estimated.")

4. The study is important but descriptive, claims of causation should be removed. For example, while you observe higher levels of CXCR5 NK cells, there are no studies show a direct regulation in the manuscript. So, claims of this regulation or causation of between inflammation should be tempered. This study would lay the groundwork for future studies exploring this direct mechanism.

We agree with the reviewer's comment. We removed all claims on causation. We tempered the discussion and added "However, while there were higher levels of CXCR5 NK cells in mesLN of SIVagm-infected AGM, our studies did not analyze if there is a direct, causative link." We also added: "This study lays the groundwork for future studies exploring direct or indirect mechanisms..." in the revised manuscript.

Minor comments:

1. For the dot plots, would be easier to see if the circles are enlarged and the error bars are removed. I would revise all graphs to be dot plots and not shift with bar or box and whisker.

Thank you for this comment. As mentioned above, we changed all the figures according to the reviewer's advices.

2. For statistical tests, I believe a nonparametric test should be used for rank such as the Wilcoxon for this number of animals. This would include the analysis of the supplementary material.

The reviewer is right, one should use a nonparametric test. This is indeed what we did, we used the nonparametric Mann-Whitney test (also called Wilcoxon-Mann-Whitney) in the figures as well as in the supplementary material. We now indicate the method used in all legends as well as in the material and method section of the revised manuscript.

3. More details about the viral infections (route of delivery etc. as well as viral loads from the blood should be shown in the results.

This information has now been added in Supplemental Table 1A and 1B. Moreover, a Figure has been added and the individual viremia levels at time of analysis are shown in Suppl. Table 1A and B and in Supplemental Fig.1A (Fig.S1A).

4. Figure 1 B should be explained in the results. Similarly, so should figure 1C. I now see 1C is after 1D, but unclear what this adds to the manuscript. Also, the figure is not labeled well.

We apologize for the confusion. We corrected the labeling of the figures and describe the results shown in all figures in the text of the revised manuscript.

5. For figure 1D, what is acute and pos? This is not explained. Also, the total number of NK cells identified should be shown here in addition to the percentage. If the SIV negative do not have many NK cells that could explain the frequency difference.

It was explained in the legend, however, we totally agree that this was not sufficient clear. "SIVAcute" meant a time point in acute infection (day 9 p.i), while "SIVPos" meant the

chronically infected animals. The latter was now changed from “SIVpos” to “SIVchronic” (see Fig.1C in the revised manuscript). We thank the reviewer for this comment. Unfortunately, we did not include truecount in the flow cytometry analyses and do not have the total numbers for the tissues. In order to take into account the reviewer’s comment, we now specify also in the main text of the revised manuscript that we only evaluated “the proportion of CXCR5+ NK cells among total NK cells”.

6. The last line of the first section of results should be revised, the authors did not show direct evidence for protection of the BCF from viral replication.

We show that in NK cell-depleted animals, this leads to increase in viral replication in the BCF of the mesLN. Although other factors are not excluded to play a role, this generally is sufficient to indicate a causative link between the cells and viral control (ie Huot et al, Nat Med 2017).

7. Again, for figure 1E, the total number of B cells would be helpful to confirm if this was a real observation or simply because of the really low levels of IgA B cells.

As explained above, we do not have the total numbers of B cells in the tissues. We do show though the frequencies of total B cells in the Suppl. Fig.S1F-G. In addition, in order to take the reviewer’s comment into consideration, we now added this limitation in the manuscript (page 6: “In the absence of total numbers, we cannot exclude that some changes were not detected or overestimated.”) and specified that we evaluated the frequencies and not the total numbers of these cells to avoid any confusion (page 6).

8. The observation of increased IgM should also be included if remain significant because of a new Wilcoxon test. Why only the IgA is the focus?

In this study we focused on IgA isotype because AGM do not develop chronic inflammation and gut disruption despite the high viremia and high virus production from gut tissues and we aimed to study IgA, because of its known anti-inflammatory functional role in gut tissue protection. Moreover, we were interested in IgG, because they are known to be increased in HIV and SIVmac infections and hypergammaglobulinemia has been associated with chronic immune activation in HIV and SIVmac infections. Also, SIVagm infection in AGM has been previously shown by others to develop higher-magnitude plasma GP120-specific IgA and IgG responses than SIVmac infection in MAC. In order to take into account, the reviewer’s suggestion, we now mention the increase of IgM, and always show the IgM values in the main Figures, whenever we have measured them (i.e. new Fig. 1E-F, Fig. 2E-F and Fig. 3C/F).

9. Figure 2A-C should be dot plots with error bars removed. Labeling should be revised and consistent with previous figures.

Thank you for this comment. We changed all the figures according to the reviewer's advices.

10. Figure 2E-F (why only MAC shown here) Compare with AGM. With revised statistics, this may not be as striking and could be moved to the SOM.

We show both MAC and AGM (in Figures 2E and 2F, respectively). In order to be clearer, we added on the figures that the changes were non-significative ("ns").

11. For figure 3 in the blood, why are the uninfected controls included. This will be important since the spread of IgA memory cells seems large.

Thank you for this comment. We now show the analyses both when analyzing uninfected and infected animals together, as it was already presented similarly in the literature with (e.g., Mudd et al, Nat Comm, 10.1038/s41467-018-05528-3). And in addition, in order to address the reviewer's comment, we also display now the data obtained when excluding the uninfected monkeys from the correlation analyses (Fig1F, Fig.S2A-C, Fig.3I-J, Fig.S5 and Fig.S6 of the revised manuscript).

12. Again, figure labeling needs to be revised for 3G-H and revised statistics.

The figures have been revised according to the reviewer's comments.

13. At the end of the figure 3 results paragraph the sentence was cut off. Starting with Although, GP or gp140 should be used.

We thank the reviewer to have notified it. We corrected the sentence and nomenclature accordingly.

14. Figure 4F/G should be removed since not significant. Could be presented in the SOM.

We removed the graphs to the Supplementary Figure 8, as suggested.

Reviewer #2 (Remarks to the Author):

Major comments:

1. Figure 1 A, B, C: can the results be quantified?

We have not quantified the viral load in the tissues in this study. This has been done in AGMs in previous studies (Huot et al, Nat Med, 10.1038/nm.4421). These and other previous studies have demonstrated a very low viral replication levels in LN of natural hosts (Brenchley et al, Blood 2012 10.1182/blood-2012-06-437608; Martinot et al, 10.1371/journal.pone.0057785). We have focused here on the spatial dynamics. We evaluated NK cells' potential to migrate into B cell follicles by analyzing the frequencies of CXCR5 expression. A recent study confirmed the causative link between CXCR5 expression on NK cells and their presence in B cell follicles in human lymph nodes during HIV-1 infection (Guo et al, EBioMedicine, doi: 10.1016/j.ebiom.2021.103794).

2. Figure 1F, Figure 3 I and J, and Figure S4 it is not appropriate to group SIV infected and uninfected animal for this type of analysis.

The concomitant use of data from infected and uninfected animals for such analyses has been done in other studies (ie Mudd et al, Nat Comm 2018 doi: 10.1038/s41467-018-05528-3). In order to reply to the reviewer's comments, we kept the same figures, but performed statistical analyses with the SIV-infected animals only as suggested by the reviewer and now report in the text of the result section also the corresponding rho and p values for data with SIV-infected animals only (highlighted in yellow in the main text of the revised manuscript) and also added new figures on the correlations with infected data only (Fig.1F, Fig.S2A-C, Fig.3I-J, Fig.S5 and Fig.S6 of the revised manuscript). We also removed from the abstract the conclusions that were obtained by grouping SIV-infected and uninfected animals.

3. What is the effect of depleting IL-15 on other cell types? Are HIV-specific CD8 T cells also impaired? How specifically is it affecting NK cells?

In our previous study by Huot et al. (Nat Med, 2017), we demonstrated that in these monkeys, the frequencies of several subsets of CD8+ T cells and CD4+ T cells were indeed modified in LN after anti-IL15 treatment during chronic SIVagm infection (Huot et al, Nat Med 2017, 10.1038/nm.4421). However, these modifications were less dramatic than for NK cells, in line with what has been reported in other studies in macaques (doi: 10.1038/nm.4421 and de Gottardi *et al*, JI 2016, doi: 10.4049/jimmunol.1600065). SIV-specific CD8 T cells were not measured, because it has been shown that SIV-specific CD8+ T cells in LN have only very little anti-SIVagm suppressive activity (doi: 10.1128/JVI.01779-08 and 10.1128/JVI.01841-08).

4. Sup Figure 3. Levels of sCD14 are high at baseline in AGM, even higher than in SIVmac. How do the authors explain this result?

Indeed, the high basal level of sCD14 were already observed in other studies in AGM (doi: 10.1371/journal.ppat.1003011; 10.1371/journal.ppat.1008333). At the opposite, cynomolgus macaques display lower basal level of sCD14 as also found in our study (doi: 10.1371/journal.ppat.1006048). Interestingly, in sooty mangabeys (another SIV natural host, which is phylogenetically closer to the *Macaca* genus than AGMs), the sCD14 basal level is close to that in cynomolgus macaques (doi: 10.1128/JVI.06813-11). This indicates species-specific differences, which are not related to the pathogenic or non-pathogenic outcome of the SIV infection.

5. Unclear what is supporting the statement on the higher affinity of IgG compared to IgA.

This was an error in wording; we changed the term in the revised manuscript and apologize.

6. The authors suggest that total IgA responses are protecting from microbial translocation but there is no increase in LBP or iFABP in SIVmac to support this concept.

The reviewer is right. We only observed an increase of sCD14, but no increase of LBP or I-FABP in the chronically infected macaques. There could be several reasons for that. It might be that the number of animals studied was too limited to see a statistically significant difference. It could also be that the animals have not yet enough progressed toward disease, due to the species (cynomolgus macaque) and virus (SIVmac251), which are known to progress generally more slowly than rhesus macaques infected with SIVmac239. It would be interesting in the future to analyze the IgA responses with regard to inflammation levels and microbial translocation in rhesus macaques infected with SIVmac239. We now discuss this issue in the discussion section of the manuscript. Of note, the increase in sCD14 was still significant and supports our conclusion. As mentioned above in response to reviewer 1, we tempered our conclusions. We removed statements that might have been suggestive of causative links and added in the discussion that our results lay the groundwork for future studies.

Minor comments:

1. Introduction: the work from Shulzhenko et al (Nat Med 2011, PMID 22101768) is also worth putting into context.

Thank you for this comment. We updated the introduction by citing the work and reference (page 4).

2. Figure 1 A and B the color coding in the legend is wrong.

We thank the reviewer for having noticed that. The color coding in the figures was correct, but we had indicated the false Figure numbers in the text. We have corrected the Figure numbers in the revised manuscript and apologize for this error.

3. How representative are the Figure 1 A, B, C, and Figure 2 D? There is no indication of how many tissue sections were analyzed. The same applies to the supplementary figures with microscopy images.

We added this information in material and method, at the end of the “Immunofluorescent staining” section. The fluorescence staining was observed and captured by a spinning-disk on three tissue sections per monkey (Immunofluorescent staining and Fluorescent In Situ Hybridization).

4. There is a sentence missing at the end of the 4th paragraph of the results. We thank the reviewer for this comment. We corrected it.

5. Figure 4 F and G, the N on the panels does not match with what is indicated in the legend. In general, the N for each panel could be better indicated in the legends.

We thank the reviewer for this comment. We corrected it.

6. Page 16, flow cytometry does not generate images. It was an error; we changed the term and thank the reviewer.

Reviewer #3 (Remarks to the Author):

Major comments:

1. In Figure 1C the authors show an increase in cells carrying viral RNA in animals depleted of NK cells by anti-IL15 treatment. Do the authors know what happens if anti-IL15 treatment is stopped? Do the viral RNA carrying cells go away - are they eliminated or do they transition to latently infected cells. This could be assessed by comparing DNA levels at different time points. It would be interesting to know if the authors are observing a role for NK cells in eliminating virus infected cells or if they are observing a pro-latency effect, as previously reported for CD8+ T cells in SIV-infected macaques by the laboratory of Guido Silvestri.

The reviewer raises several highly interesting questions. Addressing these questions could constitute an own study by itself. In the paper by Huot et al. (Nat Med 2017), it was

demonstrated that both viral RNA and viral DNA copies increased after NK cell depletion in LN from chronically SIVagm infected AGMs. In a subsequent study, it was shown that the NK cells from LN of chronically infected AGM display a strong cytotoxic activity, while cytokine production was low (doi: 10.1038/s41467-021-21402-1). Moreover, the CXCR5+ and differentiated NK cells from the LN of SIVagm-infected AGM showed an increased potential for ADCC activity. These results suggest that NK cells here rather play a role in the elimination of virus-infected (antigen-producing) cells in LN, by direct cytotoxicity and/or ADCC. This has though not directly been analyzed.

2. The authors show the presence of NK cells in the mesenteric lymph node associates with reduced cells carrying viral RNA. Is there also a reduced number of cells carrying viral DNA?

In the previous study in AGMs, we have shown that cell-associated viral DNA is also reduced in the LN (Huot 2017). Other previous studies have also shown that cell-associated viral DNA in LN of SIVagm-infected AGMs is lower when compared to HIV and SIVmac infections (Goldstein S. et al, J Virol 74,11744-11753, 2000 ; Diop O. M. et al, J. Virol 74, 7538-7547, 2000; Müller MC and Barré-Sinoussi, Front Biosc 2003; Broussard S. R. et al. J Virol 75,2262-2275, 2001).

3. Could the observation of higher NK cell numbers in the LN of AGM compared to MAC reflect a species difference upon any infection? What is seen if you infect the two species with a virus other than SIV?

This is a very interesting question. To our knowledge, there is no study with another virus that compared the NK cell numbers in LN of AGM versus Macaques. Future studies on NK cell responses in secondary lymphoid organs need to be performed. Of note, before infections, there was no significative difference regarding NK cell numbers and location within LN between the two non-human primate species (doi: 10.3389/fimmu.2020.02134).

4. What is known about Fc alpha receptor expression on NK cells in AGM? Could increase frequency of IgA drive antibody-dependent NK cell responses?

IgA driven antibody-dependent NK cell responses has been described to occur via the Fc α RI receptor (CD89). We added a new Supplemental Table (Supplemental Table 4) to describe the gene expression profile of CD89 in NK cells during SIVagm infection. The level of transcripts seemed to be present but low, when compared to transcript expressions of other genes. We previously reported that SIVagm infection in AGM is associated with induction of CXCR5+ NK cells and highly differentiated NK cells in LN that express more frequently Fc-gamma receptors (CD16, CD32, CD64) (Huot 2017, Huot 2021). These highly differentiated NK cells in the AGM LN also expressed a transcriptome profile that is reminiscent of NK cells with ADCC activity (decreased Fc ϵ R1 and increased CD3e expression; Huot et al, 2021). In general, it is known that mature NK

cells display indeed an increased ADCC activity. Thus, it is possible that the ADCC NK activity could participate in the control of SIVagm infection. However, whether this is mediated predominantly via an IgA or IgG recognition process will need to be further investigated. In order to address the reviewer's interesting question, we now added this in the discussion.

Minor comment:

1. The first paragraph on page 7 appears to be truncated.

We thank the reviewer for having notified it. We corrected it.

REVIEWERS' COMMENTS:

Reviewer #1 (Remarks to the Author):

The authors have addressed my comments and has improved the manuscript. My only lingering comment is that there is a lot of nonsignificant data presented that could be moved to supplemental data while leaving the significant different data in the main text.

Reviewer #2 (Remarks to the Author):

The manuscript is significantly improved after the first round of revisions. Some minor details still need to be addressed:

Sup Figure 5 is missing some panels in A and B.

The manuscript should be revised for typos, some example are listed below:

We thus investigated NK cells in LN mesLN

Signs of robust viral production were observed only in in mesLN

SIVagm-infected AGM

unlike to chronic SIVagm-infected animals

IFN-g

Reviewer Manuscript Questions – Second round

We sincerely thank the reviewers for having taken their time to read the revised manuscript and to have provides helpful suggestions that improves the manuscript.

Reviewer #1 (Remarks to the Author):

The authors have addressed my comments and has improved the manuscript. My only lingering comment is that there is a lot of nonsignificant data presented that could be moved to supplemental data while leaving the significant different data in the main text.

In order to follow the reviewer's suggestion, we moved the data that were mostly composed of nonsignificant data to the supplemental Figures. Thus, we moved the following Figures to the supplemental files: Figure 2 E and F; and Figure 3 A, B, C, D, and F. These figures became Figure S3 B and S3 C, and Figure S4 A-F, respectively.

We kept the graphs of Figure 1E, 2C and 4D in the main figures in order to respect a previous reviewer request to not show selected graphs but to keep the homogeneity between the figures and always show data for all three isotypes (IgA, IgM, IgG) and all compartments studied (jejunum, ileum, colon) for a given marker analyzed.

Reviewer #2 (Remarks to the Author):

The manuscript is significantly improved after the first round of revisions. Some minor details still need to be addressed:

Sup Figure 5 is missing some panels in A and B.

We thank the reviewer for this comment. Indeed, these two panels were not shown in Supp. Fig.5 because they were already presented on Principal Figure 3. In order to avoid any confusion, we changed the Supplemental Figure 5 representation, and now show them also there.

The manuscript should be revised for typos, some examples are listed below:

We thus investigated NK cells in LN mesLN

Signs of robust viral production were observed only in in mesLN

SIVagm-infected AGM

unlike to chronic SIVagm-infected animals

IFN-g

We apologize for the typos. We proofread the manuscript to revise the typo issues.

We sincerely thank the reviewers for their insightful comments which helped us to improve the manuscript. The changes made in the revised manuscript are indicate in yellow.

Reviewer #1 (Remarks to the Author):

Major comments:

1. Statistical analysis, a nonparametric rank test such as Wilcoxon-Mann-Whitney should be used for comparisons with $n=4$ or $n=6$. Ideally, correction for multiple comparisons should be done, but due to the exploratory nature of this science this should be explicitly said that these are uncorrected p values. This reanalysis may simply the conclusions and improve the conclusions of the manuscript.

The reviewer is right, one should use a nonparametric test. This is indeed what we did, we used the nonparametric Mann-Whitney test (also called Wilcoxon-Mann-Whitney). We are sorry that that was not clear. Given the low numbers in this exploratory study and generally only one parameter compared, correction for multiple comparisons is not recommended. As suggested by the reviewer, we now indicate that all values are uncorrected p values (method section of the revised manuscript).

2. Figures need to be revised to have error bars removed and dot plots. Labeling also needs to be clearer and consistent.

We thank the reviewer for this comment. We changed all the figures according to the reviewer's advices.

3. More details comparing the total number of NK cells or B cells as well as the frequency should be provided and discussed.

Several Figures in the revised manuscript (Fig. 1C and E, Fig.2 E and F, Fig.3A-F, Fig.4A-D and Suppl. Figures 1F and G) show frequencies of NK cells and B cells. We focused here on tissues (and not on blood). We did not include truecount in the flow cytometry analyses and thus unfortunately do not have the total numbers of the cells for the tissues. We mention this limitation now in the revised manuscript (pages 5 and 6: "In the absence of total numbers, we cannot exclude that some changes were not detected or under/over-estimated.")

4. The study is important but descriptive, claims of causation should be removed. For example, while you observe higher levels of CXCR5 NK cells, there are no studies show a direct regulation in the manuscript. So, claims of this regulation or causation of between inflammation should be tempered. This study would lay the groundwork for future studies exploring this direct mechanism.

We agree with the reviewer's comment. We removed all claims on causation. We tempered the discussion and added "However, while there were higher levels of CXCR5 NK cells in mesLN of SIVagm-infected AGM, our studies did not analyze if there is a direct, causative link." We also added: "This study lays the groundwork for future studies exploring direct or indirect mechanisms..." in the revised manuscript.

Minor comments:

1. For the dot plots, would be easier to see if the circles are enlarged and the error bars are removed. I would revise all graphs to be dot plots and not shift with bar or box and whisker.

Thank you for this comment. As mentioned above, we changed all the figures according to the reviewer's advices.

2. For statistical tests, I believe a nonparametric test should be used for rank such as the Wilcoxon for this number of animals. This would include the analysis of the supplementary material.

The reviewer is right, one should use a nonparametric test. This is indeed what we did, we used the nonparametric Mann-Whitney test (also called Wilcoxon-Mann-Whitney) in the figures as well as in the supplementary material. We now indicate the method used in all legends as well as in the material and method section of the revised manuscript.

3. More details about the viral infections (route of delivery etc. as well as viral loads from the blood should be shown in the results.

This information has now been added in Supplemental Table 1A and 1B. Moreover, a Figure has been added and the individual viremia levels at time of analysis are shown in Suppl. Table 1A and B and in Supplemental Fig.1A (Fig.S1A).

4. Figure 1 B should be explained in the results. Similarly, so should figure 1C. I now see 1C is after 1D, but unclear what this adds to the manuscript. Also, the figure is not labeled well.

We apologize for the confusion. We corrected the labeling of the figures and describe the results shown in all figures in the text of the revised manuscript.

5. For figure 1D, what is acute and pos? This is not explained. Also, the total number of NK cells identified should be shown here in addition to the percentage. If the SIV negative do not have many NK cells that could explain the frequency difference.

It was explained in the legend, however, we totally agree that this was not sufficient clear. "SIVAcute" meant a time point in acute infection (day 9 p.i), while "SIVPos" meant the

chronically infected animals. The latter was now changed from “SIVpos” to “SIVchronic” (see Fig.1C in the revised manuscript). We thank the reviewer for this comment. Unfortunately, we did not include truecount in the flow cytometry analyses and do not have the total numbers for the tissues. In order to take into account the reviewer’s comment, we now specify also in the main text of the revised manuscript that we only evaluated “the proportion of CXCR5+ NK cells among total NK cells”.

6. The last line of the first section of results should be revised, the authors did not show direct evidence for protection of the BCF from viral replication.

We show that in NK cell-depleted animals, this leads to increase in viral replication in the BCF of the mesLN. Although other factors are not excluded to play a role, this generally is sufficient to indicate a causative link between the cells and viral control (ie Huot et al, Nat Med 2017).

7. Again, for figure 1E, the total number of B cells would be helpful to confirm if this was a real observation or simply because of the really low levels of IgA B cells.

As explained above, we do not have the total numbers of B cells in the tissues. We do show though the frequencies of total B cells in the Suppl. Fig.S1F-G. In addition, in order to take the reviewer’s comment into consideration, we now added this limitation in the manuscript (page 6: “In the absence of total numbers, we cannot exclude that some changes were not detected or overestimated.”) and specified that we evaluated the frequencies and not the total numbers of these cells to avoid any confusion (page 6).

8. The observation of increased IgM should also be included if remain significant because of a new Wilcoxon test. Why only the IgA is the focus?

In this study we focused on IgA isotype because AGM do not develop chronic inflammation and gut disruption despite the high viremia and high virus production from gut tissues and we aimed to study IgA, because of its known anti-inflammatory functional role in gut tissue protection. Moreover, we were interested in IgG, because they are known to be increased in HIV and SIVmac infections and hypergammaglobulinemia has been associated with chronic immune activation in HIV and SIVmac infections. Also, SIVagm infection in AGM has been previously shown by others to develop higher-magnitude plasma GP120-specific IgA and IgG responses than SIVmac infection in MAC. In order to take into account, the reviewer’s suggestion, we now mention the increase of IgM, and always show the IgM values in the main Figures, whenever we have measured them (i.e. new Fig. 1E-F, Fig. 2E-F and Fig. 3C/F).

9. Figure 2A-C should be dot plots with error bars removed. Labeling should be revised and consistent with previous figures.

Thank you for this comment. We changed all the figures according to the reviewer's advices.

10. Figure 2E-F (why only MAC shown here) Compare with AGM. With revised statistics, this may not be as striking and could be moved to the SOM.

We show both MAC and AGM (in Figures 2E and 2F, respectively). In order to be clearer, we added on the figures that the changes were non-significative ("ns").

11. For figure 3 in the blood, why are the uninfected controls included. This will be important since the spread of IgA memory cells seems large.

Thank you for this comment. We now show the analyses both when analyzing uninfected and infected animals together, as it was already presented similarly in the literature with (e.g., Mudd et al, Nat Comm, 10.1038/s41467-018-05528-3). And in addition, in order to address the reviewer's comment, we also display now the data obtained when excluding the uninfected monkeys from the correlation analyses (Fig1F, Fig.S2A-C, Fig.3I-J, Fig.S5 and Fig.S6 of the revised manuscript).

12. Again, figure labeling needs to be revised for 3G-H and revised statistics.

The figures have been revised according to the reviewer's comments.

13. At the end of the figure 3 results paragraph the sentence was cut off. Starting with Although, GP or gp140 should be used.

We thank the reviewer to have notified it. We corrected the sentence and nomenclature accordingly.

14. Figure 4F/G should be removed since not significant. Could be presented in the SOM.

We removed the graphs to the Supplementary Figure 8, as suggested.

Reviewer #2 (Remarks to the Author):

Major comments:

1. Figure 1 A, B, C: can the results be quantified?

We have not quantified the viral load in the tissues in this study. This has been done in AGMs in previous studies (Huot et al, Nat Med, 10.1038/nm.4421). These and other previous studies have demonstrated a very low viral replication levels in LN of natural hosts (Brenchley et al, Blood 2012 10.1182/blood-2012-06-437608; Martinot et al, 10.1371/journal.pone.0057785). We have focused here on the spatial dynamics. We evaluated NK cells' potential to migrate into B cell follicles by analyzing the frequencies of CXCR5 expression. A recent study confirmed the causative link between CXCR5 expression on NK cells and their presence in B cell follicles in human lymph nodes during HIV-1 infection (Guo et al, EBioMedicine, doi: 10.1016/j.ebiom.2021.103794).

2. Figure 1F, Figure 3 I and J, and Figure S4 it is not appropriate to group SIV infected and uninfected animal for this type of analysis.

The concomitant use of data from infected and uninfected animals for such analyses has been done in other studies (ie Mudd et al, Nat Comm 2018 doi: 10.1038/s41467-018-05528-3). In order to reply to the reviewer's comments, we kept the same figures, but performed statistical analyses with the SIV-infected animals only as suggested by the reviewer and now report in the text of the result section also the corresponding rho and p values for data with SIV-infected animals only (highlighted in yellow in the main text of the revised manuscript) and also added new figures on the correlations with infected data only (Fig.1F, Fig.S2A-C, Fig.3I-J, Fig.S5 and Fig.S6 of the revised manuscript). We also removed from the abstract the conclusions that were obtained by grouping SIV-infected and uninfected animals.

3. What is the effect of depleting IL-15 on other cell types? Are HIV-specific CD8 T cells also impaired? How specifically is it affecting NK cells?

In our previous study by Huot et al. (Nat Med, 2017), we demonstrated that in these monkeys, the frequencies of several subsets of CD8+ T cells and CD4+ T cells were indeed modified in LN after anti-IL15 treatment during chronic SIVagm infection (Huot et al, Nat Med 2017, 10.1038/nm.4421). However, these modifications were less dramatic than for NK cells, in line with what has been reported in other studies in macaques (doi: 10.1038/nm.4421 and de Gottardi *et al*, JI 2016, doi: 10.4049/jimmunol.1600065). SIV-specific CD8 T cells were not measured, because it has been shown that SIV-specific CD8+ T cells in LN have only very little anti-SIVagm suppressive activity (doi: 10.1128/JVI.01779-08 and 10.1128/JVI.01841-08).

4. Sup Figure 3. Levels of sCD14 are high at baseline in AGM, even higher than in SIVmac. How do the authors explain this result?

Indeed, the high basal level of sCD14 were already observed in other studies in AGM (doi: 10.1371/journal.ppat.1003011; 10.1371/journal.ppat.1008333). At the opposite, cynomolgus macaques display lower basal level of sCD14 as also found in our study (doi: 10.1371/journal.ppat.1006048). Interestingly, in sooty mangabeys (another SIV natural host, which is phylogenetically closer to the *Macaca* genus than AGMs), the sCD14 basal level is close to that in cynomolgus macaques (doi: 10.1128/JVI.06813-11). This indicates species-specific differences, which are not related to the pathogenic or non-pathogenic outcome of the SIV infection.

5. Unclear what is supporting the statement on the higher affinity of IgG compared to IgA.

This was an error in wording; we changed the term in the revised manuscript and apologize.

6. The authors suggest that total IgA responses are protecting from microbial translocation but there is no increase in LBP or iFABP in SIVmac to support this concept.

The reviewer is right. We only observed an increase of sCD14, but no increase of LBP or I-FABP in the chronically infected macaques. There could be several reasons for that. It might be that the number of animals studied was too limited to see a statistically significant difference. It could also be that the animals have not yet enough progressed toward disease, due to the species (cynomolgus macaque) and virus (SIVmac251), which are known to progress generally more slowly than rhesus macaques infected with SIVmac239. It would be interesting in the future to analyze the IgA responses with regard to inflammation levels and microbial translocation in rhesus macaques infected with SIVmac239. We now discuss this issue in the discussion section of the manuscript. Of note, the increase in sCD14 was still significant and supports our conclusion. As mentioned above in response to reviewer 1, we tempered our conclusions. We removed statements that might have been suggestive of causative links and added in the discussion that our results lay the groundwork for future studies.

Minor comments:

1. Introduction: the work from Shulzhenko et al (Nat Med 2011, PMID 22101768) is also worth putting into context.

Thank you for this comment. We updated the introduction by citing the work and reference (page 4).

2. Figure 1 A and B the color coding in the legend is wrong.

We thank the reviewer for having noticed that. The color coding in the figures was correct, but we had indicated the false Figure numbers in the text. We have corrected the Figure numbers in the revised manuscript and apologize for this error.

3. How representative are the Figure 1 A, B, C, and Figure 2 D? There is no indication of how many tissue sections were analyzed. The same applies to the supplementary figures with microscopy images.

We added this information in material and method, at the end of the “Immunofluorescent staining” section. The fluorescence staining was observed and captured by a spinning-disk on three tissue sections per monkey (Immunofluorescent staining and Fluorescent In Situ Hybridization).

4. There is a sentence missing at the end of the 4th paragraph of the results.
We thank the reviewer for this comment. We corrected it.

5. Figure 4 F and G, the N on the panels does not match with what is indicated in the legend. In general, the N for each panel could be better indicated in the legends.

We thank the reviewer for this comment. We corrected it.

6. Page 16, flow cytometry does not generate images.
It was an error; we changed the term and thank the reviewer.

Reviewer #3 (Remarks to the Author):

Major comments:

1. In Figure 1C the authors show an increase in cells carrying viral RNA in animals depleted of NK cells by anti-IL15 treatment. Do the authors know what happens if anti-IL15 treatment is stopped? Do the viral RNA carrying cells go away - are they eliminated or do they transition to latently infected cells. This could be assessed by comparing DNA levels at different time points. It would be interesting to know if the authors are observing a role for NK cells in eliminating virus infected cells or if they are observing a pro-latency effect, as previously reported for CD8+ T cells in SIV-infected macaques by the laboratory of Guido Silvestri.

The reviewer raises several highly interesting questions. Addressing these questions could constitute an own study by itself. In the paper by Huot et al. (Nat Med 2017), it was

demonstrated that both viral RNA and viral DNA copies increased after NK cell depletion in LN from chronically SIVagm infected AGMs. In a subsequent study, it was shown that the NK cells from LN of chronically infected AGM display a strong cytotoxic activity, while cytokine production was low (doi: 10.1038/s41467-021-21402-1). Moreover, the CXCR5+ and differentiated NK cells from the LN of SIVagm-infected AGM showed an increased potential for ADCC activity. These results suggest that NK cells here rather play a role in the elimination of virus-infected (antigen-producing) cells in LN, by direct cytotoxicity and/or ADCC. This has though not directly been analyzed.

2. The authors show the presence of NK cells in the mesenteric lymph node associates with reduced cells carrying viral RNA. Is there also a reduced number of cells carrying viral DNA?

In the previous study in AGMs, we have shown that cell-associated viral DNA is also reduced in the LN (Huot 2017). Other previous studies have also shown that cell-associated viral DNA in LN of SIVagm-infected AGMs is lower when compared to HIV and SIVmac infections (Goldstein S. et al, J Virol 74,11744-11753, 2000 ; Diop O. M. et al, J. Virol 74, 7538-7547, 2000; Müller MC and Barré-Sinoussi, Front Biosc 2003; Broussard S. R. et al. J Virol 75,2262-2275, 2001).

3. Could the observation of higher NK cell numbers in the LN of AGM compared to MAC reflect a species difference upon any infection? What is seen if you infect the two species with a virus other than SIV?

This is a very interesting question. To our knowledge, there is no study with another virus that compared the NK cell numbers in LN of AGM versus Macaques. Future studies on NK cell responses in secondary lymphoid organs need to be performed. Of note, before infections, there was no significative difference regarding NK cell numbers and location within LN between the two non-human primate species (doi: 10.3389/fimmu.2020.02134).

4. What is known about Fc alpha receptor expression on NK cells in AGM? Could increase frequency of IgA drive antibody-dependent NK cell responses?

IgA driven antibody-dependent NK cell responses has been described to occur via the Fc α RI receptor (CD89). We added a new Supplemental Table (Supplemental Table 4) to describe the gene expression profile of CD89 in NK cells during SIVagm infection. The level of transcripts seemed to be present but low, when compared to transcript expressions of other genes. We previously reported that SIVagm infection in AGM is associated with induction of CXCR5+ NK cells and highly differentiated NK cells in LN that express more frequently Fc-gamma receptors (CD16, CD32, CD64) (Huot 2017, Huot 2021). These highly differentiated NK cells in the AGM LN also expressed a transcriptome profile that is reminiscent of NK cells with ADCC activity (decreased Fc ϵ R1 and increased CD3e expression; Huot et al, 2021). In general, it is known that mature NK

cells display indeed an increased ADCC activity. Thus, it is possible that the ADCC NK activity could participate in the control of SIVagm infection. However, whether this is mediated predominantly via an IgA or IgG recognition process will need to be further investigated. In order to address the reviewer's interesting question, we now added this in the discussion.

Minor comment:

1. The first paragraph on page 7 appears to be truncated.

We thank the reviewer for having notified it. We corrected it.